# Significant climate impacts of aerosol changes driven by growth in energy use and advances in emissions control technology

Alcide Zhao[1*], Massimo A. Bollasina[1], Monica Crippa[2] and David S. Stevenson[1]

[1] School of GeoSciences, University of Edinburgh, Edinburgh, UK

[2] European Commission, Joint Research Centre (JRC), Ispra, Italy

[*] Correspondence to: Alcide Zhao (alcide.zhao@ed.ac.uk)

**Abstract.** Anthropogenic aerosols have increased significantly since the industrial revolution, driven largely by growth in emissions from energy use in sectors including power generation, industry, and transport. Advances in emission control technologies since around 1970, however, have partially counteracted emissions increases from the above sectors. Using the fully-coupled Community Earth System Model, we quantify the effective radiative forcing (ERF) and climate response to 1970-2010 aerosol changes associated with the above two policy-relevant emission drivers. Emissions from energy use growth generate a global mean aerosol ERF (mean ± one standard deviation) of -0.31 0.22 W m$^{-2}$ and result in a global mean cooling (-0.35±0.17 K) and a precipitation reduction (-0.03±0.02 mm day$^{-1}$). By contrast, the avoided emissions from advances in emission control technology, which benefit air quality, generate a global mean ERF of +0.21±0.23 W m$^{-2}$, a global warming of +0.10±0.13 K and global mean precipitation increase of +0.01±0.02 mm day$^{-1}$. Despite the relatively small changes in global mean precipitation, these two emission drivers have profound impacts at regional scales, in particular over Asia and Europe. The total net aerosol impacts on climate are dominated by energy use growth, from Asia in particular. However, technology advances outweigh energy use growth over Europe and North America. Various non-linear processes are involved along the pathway from aerosol/precursor emissions to radiative forcing and ultimately to climate responses, suggesting that the diagnosed aerosol forcing and effects must be interpreted in the context of experiment designs. Further, the temperature response per unit aerosol ERF varies significantly across many factors, including location and magnitude of emission changes, implying that ERF, and the related metrics, need to be used very carefully for aerosols. Future aerosol-related emission pathways have large temporal and spatial uncertainties; our findings provide useful information for both assessing and interpreting such uncertainties, and may help inform future climate change impact reduction strategies.

## 1 Introduction

Climate change is driven by changes in a combination of natural and anthropogenic factors (Stocker et al., 2013). The increasing atmospheric abundance of greenhouse gases (GHGs) associated with human activities has long been recognized as the major driver of global warming since the industrial revolution. Anthropogenic emissions of aerosols and their precursor gases have also led to significant climate impacts (Boucher et al., 2013), in addition to their detrimental impacts on atmospheric visibility, human health and ecosystems. Aerosols can influence climate by absorbing/scattering shortwave radiation (aerosol-radiation interactions; Haywood and Ramaswamy (1998)), and by modifying cloud microphysics and precipitation processes (aerosol-cloud interactions; Fan et al. (2016)). Overall, anthropogenic aerosols cause a net cooling of the Earth; almost a third of the warming from increases in GHGs is thought to have been counteracted by cooling due to increased anthropogenic aerosols since the 1950s (Stocker et al., 2013). Yet, despite extensive research in the last decade that has led to significant progress in our understanding of the effects of aerosols (Ming and Ramaswamy, 2009; Shindell and Faluvegi, 2009; Allen and Sherwood, 2011; Bollasina et al., 2011; Ming and Ramaswamy, 2011; Ming et al., 2011; Boucher et al., 2013; Hwang et al., 2013; Wilcox et al., 2013; Xie et al., 2013; Shindell, 2014; Wang et al., 2015), there are still major uncertainties associated with their impacts on climate (Carslaw et al., 2013a; Fan et al., 2016; Lee et al., 2016; Fletcher et al., 2018).

In fact, aerosol forcing remains the dominant uncertainty in current estimates of radiative forcing on climate since pre-industrial times (Myhre et al., 2013). This is because of compounding uncertainties associated with the large spatial and temporal variability of aerosols, their short lifetimes, their diverse physical and chemical properties, and complex interactions with radiation and microphysical processes (Boucher et al., 2013; Carslaw et al., 2013b; Fan et al., 2016). For example, the sign and magnitude of the effect of aerosols on clouds and the intertwined effects on precipitation can vary substantially depending on emission locations, aerosol species, as well as meteorological conditions (Rosenfeld et al., 2008; Stevens and Feingold, 2009; Yu et al., 2014; Malavelle et al., 2017; Kasoar et al., 2018). Also, there are large uncertainties due to the incomplete knowledge of both historical aerosol changes and how they will evolve in the future (Gidden et al., 2018). All these uncertainties make it challenging to project future climate and to quantify the associated impacts on a range of sectors. More importantly, despite ongoing debates as to whether aerosol has larger impacts on mean climate and climate extremes compared to GHGs (Feichter et al., 2004; Xie et al., 2013; Wilcox et al., 2018), a large body of studies indicate that, per unit of forcing/warming, aerosols have significantly larger impacts than GHGs on both global mean climate (Hansen et al., 2005; Shindell, 2014; Shindell et al., 2015), as well as global/regional climate extremes (Perkins, 2015; Xu et al., 2015; Lin et al., 2016; Wang et al., 2016; Samset et al., 2018a; Zhao et al., 2018; Zhao et al., 2019).

Emissions of anthropogenic aerosols (and their precursors) have followed opposite trends between developed (decreases) and developing (increases) countries during the past few decades. For example, emissions of $SO_2$ from Asia increased steadily since the 1950s, while emissions from Europe and North America started to decline after the 1970s (Smith et al., 2011; Wang et al., 2015; Crippa et al., 2016). The decline of air pollutant emissions in Europe and North America dates back to around 1970 when the first air quality directives were implemented at the continental scale (Crippa et al., 2016). By comparison, only after about 2010 have some developing countries started to take mitigation measures. For example, Chinese $SO_2$ emissions have shown a noticeable decline since around 2012 (Silver et al., 2018; Zheng et al., 2018). As a result, India has recently overtaken China as the largest present-day emitter of $SO_2$ (Li et al., 2017). Anthropogenic aerosol-related emissions are expected to be further significantly reduced worldwide during the 21$^{st}$ century (Markandya et al., 2018). Aerosol mitigation, however, may lead to adverse climate impacts, such as the increased risk of climate extremes (Kloster et al., 2010; Samset et al., 2018a; Zhao et al., 2018; Zhao et al., 2019). A number of equally-plausible future emission pathways have been designed to seek a compromise between the impacts of air pollution on environment and climate following aerosol abatement in the near-, medium-, and long-term (Gidden et al., 2018). The uncertainty in the emission pathway alone represents a key limiting factor to a robust quantification and isolation of the overall aerosol impact on climate. Yet, possible differences in the climate response to varying aerosol/precursor emissions trajectories, all the other forcings being the same, have been mostly overlooked so far (e.g., Sillmann et al. (2013); Pendergrass et al. (2015); Bartlett et al. (2016)). This, nevertheless, is useful for partially assessing the uncertainty range of future climate projections related to uncertainties in aerosol-related emission pathways alone, despite the fact that emissions of GHGs also differ between those emission pathways.

Emission changes described above are primarily associated with three important and largely regulated sectors (industry, power generation and transport), while the residual contribution to emissions from residential and agricultural sectors is relatively stationary in time (Crippa et al., 2016; Hoesly et al., 2018). Also, such changes originate primarily from two competing emission drivers: economic growth and policy-driven emission controls (Crippa et al., 2016). The former is associated with energy use growth within the three sectors described above, while the latter includes both air pollution abatement measures and technology advances (hereinafter technology advances for short). To quantify the impacts of these factors, Crippa et al. (2016) developed the Emission Database for Global Atmospheric Research (EDGAR) retrospective air pollution emission scenarios for the period 1970-2010 (Sect. 2.1). Using a chemistry-climate model, Turnock et al. (2016) reported that the avoided aerosol-related emissions due to legislation and technology measures have improved air quality and human health over Europe, but have also led to a regional warming of up to 0.45 ±0.11 °C.

As discussed above, energy use growth and technology advances are two of the major policy-relevant drivers of past aerosol changes via, for example, changes in power generation, industry and transport. These drivers are very likely to continue to play important but competing roles in modulating future emissions of aerosols and their precursor gases, as we gradually transit to a new energy structure. An analysis of the climate impact to recent changes in the two above emissions drivers is therefore critically important for future aerosol-related climate projections and climate change impact reduction strategies. Here we perform time-slice model simulations using the fully-coupled Community Earth System Model (CESM1), seeking to quantify the climate forcing and impacts of aerosol changes related to the above policy-relevant emission drivers (energy use growth and technology advances) at both global and regional scales. The aerosol scenarios used here represent the best estimate of past emissions. Therefore, compared to idealized experiments where aerosol emissions/concentrations are scaled rather arbitrarily, the implications of this work can be more informative for future decision-making. The EDGAR scenarios, CESM1 model overview and experiment design, as well as analysis methods, are introduced in Section 2, Section 3 presents the results followed by a discussion in Section 4, and a summary in Section 5.

## 2. Emission scenarios and model experiments

### 2.1 The EDGAR retrospective air pollution scenarios

Based on the EDGAR4.3.1 best estimate for 1970 and 2010 (REF2010), the EDGAR retrospective emission scenarios were designed to quantify the effectiveness of 1970-2010 changes in energy use and efficiency, technology progress and end-of-pipe emission reduction measures (Crippa et al., 2016). For the 1970-2010 changes in emissions of each individual aerosol/precursor species, please refer to Fig. S1 in the supplementary file, as well as Crippa et al. (2016) for more details. These retrospective scenarios focus on sectors including power generation, industry and road transport (the most regulated sectors), whereas emissions from all other sectors are the same as those in REF2010. The highest emission scenario (STAG_TECH, Table 1) assumes no further improvements in technology and abatement measures after 1970, but energy use and different fuel mix as in REF2010. The second and lowest emission scenario (STAG_ENE) assumes stagnation of energy consumption since 1970, while fuel mix, energy efficiency, emission factors and abatements are the same as REF2010. Therefore, the difference between REF2010 and STAG_TECH represents the 2010-1970 emission reductions due to technology advances. Similarly, the difference between REF2010 and STAG_ENE represents the 2010-1970 emission increase due to energy use growth. Note carefully that the retrospective emission scenarios were deliberately designed to have emission changes from these two competing drivers to not add up to those of the total 1970-2010 changes, for the aim of quantifying the associated impacts from a what-if perspective. For example, what would be expected assuming that we had not introduced any emission control technologies since the 1970s? For more details regarding the nonlinearity associated with the retrospective emission scenarios, please refer to Crippa et al. (2016).

### 2.2 Model and experiment design

We carry out time-slice simulations (Table 1) using the fully-coupled Community Earth System Model (CESM1; Hurrell et al. (2013)) at the nominal 1-degree resolution. The motivation of carrying out time-slice model simulation for this particular period has been justified in Zhao et al. (2019b). The atmosphere component of CESM1 is the Community Atmosphere Model 5 (CAM5) in which surface concentrations of $CO_2$ and $CH_4$ are prescribed with seasonal cycles and latitude-gradients (Conley et al., 2012). CAM5 includes a three-mode (Aitken, accumulation, and coarse) aerosol scheme (Modal Aerosol Mode 3; MAM3). Several aerosol species (sulphate, organic carbon (OC), black carbon (BC), sea-salt, and dust) are simulated and their number concentration and mass are prognostically calculated for each aerosol mode. Simple gas-phase chemistry is included for sulphate species: $SO_2$ is converted into $SO_4$ through both gas-phase OH oxidation and aqueous-phase oxidation by $H_2O_2$ and $O_3$ (Liu et al., 2015a; Liu et al., 2015b; Tilmes et al., 2015). Note that CESM1 (CAM5) has a relatively larger aerosol

forcing compared to other Coupled Model Intercomparison Project Phase 5 (CMIP5) models, likely due to the large cloud adjustments through cloud water path in MAM3 (Allen and Ajoku, 2016; Malavelle et al., 2017; Zhou and Penner, 2017). In light of this and considering the overall uncertainties in the representation of aerosol effects, we underscore that all results and discussions below should be interpreted in the context of CESM1-CAM5.

Long-lived GHGs, natural aerosols and other reactive gases emissions/concentrations are obtained from Lamarque et al. (2011) for 2010. Ozone concentrations for 2010 are from the Whole Atmosphere Community Climate Model (WACCM) simulations (Marsh et al., 2013). Anthropogenic aerosols and their precursor emissions are from the EDGAR retrospective scenarios (Sect. 2.1). The EDGAR emission sectors are remapped to conform to CAM5 emissions following Lamarque et al. (2010).

The baseline 2010 experiment (B10) was initialised using the year 2010 model dump from one ensemble member (No 34) of

the CESM1 large ensemble (Kay et al. (2015) transient historical experiment, and was driven by the 2010 all forcing factors (Table 1). Also, we have three perturbation experiments where anthropogenic aerosols are perturbed using different emission scenarios (i.e., the 1970 best estimate, STAG_ENE and STA_TECH as described in Sect. 2.1) while all others forcing agents (e.g., GHGs, natural aerosols, land use, solar forcing) are the same as in the B10 run, in order to differentiate the impacts of the two emission drivers (refer Table 1 for more details). For each case, we have a paired set of simulations: one with sea

surface temperature and sea ice fixed (hereinafter Fsst), and the other with a fully coupled ocean (Fcpd). All Fcpd simulations were integrated to equilibrium (i.e., where the surface climate system equilibrates to imposed perturbations; NB the deep ocean may take longer to equilibrate) after the initial perturbation, with repeated annual cycles of the forcings. For example, the baseline B10 simulation was integrated into equilibrium under constant 2010 forcings. Note carefully that the length of each integration is different, and is deemed sufficient for analysis once the top-of-the-atmosphere radiation imbalance does not

show significant trends any more during the last few decades of each run (stabilizing at values around ~0.3 W m$^{-2}$), following recent works (Samset et al., 2016; Myhre et al., 2017; Samset et al., 2018b). We analyse the last 30 years of each equilibrium simulation and show differences between the baseline and perturbed simulations. Specifically, we denote 'Best Estimate' as the response to the best estimate of 1970-2010 total net anthropogenic aerosol-related emissions, 'energy use growth' as the response to emissions increases due to growth in energy use, and 'technology advances' as the response to avoided emissions

from advances in emission control technology. Changes in anthropogenic aerosol/precursor emissions associated with the two emission drivers are presented in Fig. S1. The statistical significance of the difference between each two (baseline and perturbed) sets of 30-yr model runs is estimated by the two-sided student t-test (p-value <0.05), and accounting for serial autocorrelation by adjusting the degrees of freedom following (Nychka et al., 2000).

The paired Fsst simulation is under the same forcings as the corresponding Fcpd simulation, and are integrated for 40 years

from the initial condition. The last 30-years of each Fsst simulation is used to diagnose the effective radiative forcing (ERF) at the top-of-the-atmosphere (top of the model in this case, ~3.6 hPa) following Forster et al. (2016). Additionally, we carried out similar Fsst simulations to diagnose the ERFs of the best estimate of 1970-2010 changes in the three major anthropogenic aerosol species (BC, OC and sulphate species; Fig. S1a-c). For example, we have a perturbation Fsst run in which only emissions of sulphate species are changed back to 1970 levels while all other forcings are the same as B10 to diagnose the

ERF due to 1970-2010 sulphate aerosol changes.

## 3 Effective radiative forcing and climate responses

### 3.1 Effective radiative forcing

Figure 1 shows changes in aerosol burdens and the diagnosed ERF associated with the best estimate of 1970-2010 changes of BC, OC and sulphate species (Fig. S1a-c). It can be seen that changes in the burdens of all aerosol species are statistically

significant almost worldwide, while areas with statistically significant ERFs are very confined. Aerosol burdens display opposite changes between Asia and industrialized regions of Europe and North America. For instance, the burden of SO$_4$

increases by 5.6 mg m$^{-2}$ in Asia but decreases by -4.0 mg m$^{-2}$ in Europe. BC changes are shown to generate a global mean positive radiative forcing of +0.06 W m$^{-2}$; the spatial pattern of BC ERF is positively correlated to that of the burdens, resulting in peak values over Asia and Africa. In contrast to BC, OC changes generate a global mean negative forcing of -0.04 W m$^{-2}$; note also the general spatial anti-correlation between OC burdens and ERFs. The global mean ERF of sulphate aerosol changes is small and positive, because of the partial cancellation between the negative forcing from sulphate aerosol increases over Asia and the pronounced positive forcing from sulphate aerosol reductions over Europe and North America which is amplified over the Arctic (Fig. 1f). Regional ERF values are dominated by the 1970-2010 changes in sulphate species. It is worth noticing that the individual ERF values of each aerosol species do not add up to that due to the simultaneous changes in all these at the global scale (Fig. 2d). A further discussion on this is provided in Sect. 4.1.

The spatial patterns of the changes in the 550-nm Aerosol Optical Depth (AOD) are strongly correlated with those of aerosol burdens (compare Fig. 1a-c to Fig. 2a). Therefore, instead of aerosol burdens, we turn to change in the total AOD of all aerosol species for the three scenario experiments where all aerosol species change simultaneously. The total net 1970-2010 AOD changes (Fig.2a), not surprisingly, display a sharp contrast between Asia (+0.036) and Europe (-0.023) and North America (-0.004). This, as described above, is mainly driven by changes in sulphate aerosols (Fig. 1c). The 1970-2010 aerosol-related emission changes produce a global mean ERF of -0.11±0.14 W m$^{-2}$, with marked regional values over Europe (+2.3±1.4 W m$^{-2}$) and Asia (-1.06±0.72 W m$^{-2}$; Fig.3b). Aerosol changes due to energy use growth lead AOD to increase almost worldwide (Fig.2b), resulting in a global mean ERF of -0.31±0.21 W m$^{-2}$, with the most noticeable negative forcing of -0.88±0.60 W m$^{-2}$ over Asia followed by -0.51±0.53 W m$^{-2}$ over North America (Fig. 3b). By contrast, the avoided emissions due to technology advances lead AOD to decrease predominately over the Northern Hemisphere (Fig. 2c), and generate a global mean positive forcing of +0.21±0.23 W m$^{-2}$ (Fig. 2f). The most noticeable changes are found over Europe (+1.16±1.11 W m$^{-2}$) and North America (+0.53±0.49 W m$^{-2}$). It worth noting the AOD increases over Southern Africa are due to increases in sea-salt and OC which may be related to the additional warming induced changes in meteorology in the technology advances experiment.

## 3.2 Temperature responses

Figure 4 shows the spatial distribution and zonal mean profile of the surface air temperature responses. Also see Fig. 3c for the regional mean values. It can be seen that the majority of statistically significant temperature changes in response to aerosol changes are over the ocean rather than the land. This is particularly true for the energy use experiment, and may reflect the fact that the equilibrium climate response is dominated by the slow response of the ocean. In response to the 1970-2010 aerosol changes, the global mean surface temperature changes by -0.26±0.14 K, while there are confined and weak warming patterns over local regions including Eastern Europe and USA (Fig. 4a). The zonal mean temperature changes show significant cooling of the Northern Hemisphere that is amplified over the Arctic (-0.83±0.60 K), together with a less pronounced cooling in the Southern Hemisphere. The sign of global mean surface temperature change due to 1970-2010 aerosol changes is consistent with that of ERF. Note, however, the inconsistency between regional mean ERFs and temperature responses (e.g., particularly over the Arctic, Europe and the Southern Ocean). Further analysis shows such inconsistency may be associated with reductions in Arctic clouds due to a widespread low-tropospheric anomalous anticyclone over the Arctic together with an extensive cyclonic circulation centred over Central Europe (Fig. S2a), as well as the resultant decreases in surface net radiation ((Fig. S2b). The anomalous southerlies transport cold air southward (Fig. S1a) and partially oppose the warming associated with the local positive ERF in high latitudes, leading to pronounced high-latitude cooling that is further amplified through the sea ice-albedo-cloud related feedbacks. (Kay et al., 2012; Najafi et al., 2015; Sand et al., 2015; Navarro et al., 2016; Dobricic et al., 2019). For more details, please refer to Zhao et al. (2019b).

Aerosol-related emission changes from energy use growth result in a more prominent cooling that is statistically significant almost worldwide, and over the oceans in particular (Fig. 4b), with a global mean cooling of -0.35±0.17 K. The cooling is enhanced over the Arctic (-0.92±0.73 K). The zonal mean temperature response displays significant cooling across all latitude

bands, with peak values found at the North Pole (up to -1.5 K). It can be seen that both the spatial pattern and zonal mean of temperature changes due to energy use growth induced aerosol changes resembles very well those of the 1970-2010 Best Estimate, but with much larger magnitudes of changes. This demonstrates energy use as a major contributor to the climate impacts induced by the 1970-2010 aerosol changes, which is particularly the case over Asia and the Arctic. Changes in sea level pressure and low-level circulation (Fig. S2c) show a further enhanced anomalous Arctic high and anomalous high latitudinal southerlies compared to the Best Estimate experiment. This seems to be reinforcing the cooling effect due to decreases in surface net radiation as aerosols increase (Fig. S2d), and may explain why the surface cooling is more prominent compared to that in the Best Estimate experiment.

The avoided aerosol-related emissions from technology advances (Fig. 2c) lead the globe to warm by +0.10±0.13 K, with the most pronounced responses over the Arctic (+0.22±0.61 K) and North America (+0.18±0.19 K). Yet, the warming effects can also be seen over other regions including Asia, Africa, and South America, despite the relatively smaller aerosol reductions in these regions related to technology advances. Note that all the temperature responses have large uncertainties. The zonal mean temperature response is only distinguishable from zero over the Northern Hemisphere mid-latitudes (~30°N) and the polar regions. There is a noticeable cooling pattern over Europe despite the large positive forcing (+1.16 W m$^{-2}$; Fig. 2f). This seems to be related to adjustments in the atmospheric circulation that brings cold air from higher latitudes (Fig. S2e), overwhelming the effects of local surface radiation increases (Fig. S2f).

The competition between technology advances (Fig. 4c) and energy use (Fig. 4b) growth can be clearly seen in modulating the spatial pattern of global temperature changes, with the global mean values (0.10 K and -0.35 K respectively) almost adding up to that in the Best Estimate experiment (-0.26 K). Note however that regional mean values do not add up, with Northeast Eurasia and the Atlantic Ocean in particular, where the two drivers reinforce each other in changing local temperature changes. As described above, despite the broad consistency between the patterns of aerosol ERF (Fig. 2) and temperature response (Fig. 4), there are also notable dissimilarities. This is particularly true over the high latitudes, where changes in atmospheric circulation may play important roles in local surface temperature responses. To point this out more clearly, we calculate the temperature response per unit aerosol ERF (temperature sensitivity) over various domains (Fig. 5). It can be seen that the relationship between ERF and temperature response is far from being linear even at the global scale and over latitudinal bands. For example, the global mean sensitivity value ranges from 0.5 K (W m$^{-2}$)$^{-1}$ in the technology advances experiment to 2.4 K (W m$^{-2}$)$^{-1}$ in the Best Estimate experiment. Also, note the negative temperature sensitivity values for various regions (e.g., the Arctic, Europe, North America, and Asia), questioning ERF as a useful predictor of temperature change for aerosols.

### 3.3 Precipitation responses

Changes in precipitation show complex spatial patterns (Fig. 6) and much larger uncertainties (Fig. 3d) compared to temperature responses. Overall, the 1970-2010 aerosol changes result in a global precipitation reduction (-0.04±0.02 mm day$^{-1}$), with the most pronounced changes over Asia (-0.13±0.28 mm day$^{-1}$) and adjoining oceans (Fig. 6a). By comparison, precipitation increase can be seen over Europe (+0.03±0.08 mm day$^{-1}$) and the North Atlantic Ocean. Despite the large uncertainties, the zonal mean changes show precipitation reductions at almost all latitude bands. The pronounced precipitation reductions over Asia reflect partly the 20$^{th}$ century drying trend of the Asian monsoon (Yihui and Chan, 2005; Lau and Kim, 2006; Bollasina et al., 2011; Ganguly et al., 2012; Polson et al., 2014; Song et al., 2014; Lau and Kim, 2017; Ma et al., 2017), as supported by changes in the low-level circulation patterns (i.e., the prominent anomalous easterlies over the Northern Indian Ocean that weaken the South Asian monsoon, as well as the anomalous cyclonic circulations over the tropical Western Pacific Ocean that weaken the East Asian monsoon; Fig. S2a).

The globe, especially land areas, gets even drier in response to aerosol changes from energy use growth (Fig. 6b). The precipitation change in Asia (-0.11±0.30 mm day$^{-1}$) is close to that associated with the best estimate of 1970-2010 aerosol changes (-0.13±0.28 mm day$^{-1}$). This, as also in temperature response, suggests that aerosol changes from energy use growth

exert the predominant control on precipitation changes over Asia. The precipitation reduction is also notable over Europe (-0.05±0.09 mm day$^{-1}$). Along with precipitation decreases at almost all latitude bands, and the tropics in particular, zonal mean precipitation changes show a weak but southward shift of the Inter-Tropical Convergence Zone (ITCZ), leading to weak precipitation increases over the Southern Hemisphere subtropics (10-30°S). As in the Best Estimate case, changes in low-level circulation (Fig. S2c) also suggest a weakening of the South Asian monsoon. This seems to be consistent with the relatively less prominent precipitation reduction over South Asia and the North Indian Ocean (Fig. 6b). In contrast, the anomalous cyclonic circulation over the tropical Western Pacific is enhanced, leading to further precipitation decreases over East Asia but increases over the adjacent ocean.

On the contrary to energy use growth, technology advances lead precipitation to increase globally (+0.01±0.02 mm day$^{-1}$) and particularly in the Northern Hemisphere, along with its warming effect (Fig. 4c). There are precipitation increases over Europe, Mediterranean and Northern Africa (Fig. 6c), along with the anomalous cyclonic circulation that brings moisture from the Atlantic eastward (Fig. S2e), as aerosol reductions result in more radiation reaching the land surface (Fig. S2f). In comparison, prominent precipitation decreases can still be seen over Southeast Asia and the North Indian Ocean, likely due to the low-level circulation anomalies (i.e., anomalous easterlies over the North Indian Ocean and westerlies over the Equatorial Pacific Ocean) that impede the climatological moisture transport. Meanwhile, the zonal mean precipitation profile shows a marked northward shift of the ITCZ with notable precipitation reductions over the Southern Hemisphere tropics.

Similar to temperature changes, the 1970-2010 precipitation changes induced by aerosol changes also demonstrate competition between the two emission drivers, yet the values do not add up to that in the Best Estimate case even when globally averaged. Generally, the global mean precipitation changes with temperature at a rate of 0.09-0.15 mm day$^{-1}$ K$^{-1}$. This is slightly larger than the multi-model mean estimate (~28.6 mm yr$^{-1}$ K$^{-1}$, i.e., ~0.08 mm day$^{-1}$ K$^{-1}$) for the slow climate response component derived from the Precipitation Driver Response Model Intercomparison Project (PDRMIP; Samset et al. (2016)). Most of the global and regional mean responses follow to some extent the linear increase (compare Fig. 3c to 3d), but Asia, Europe and the Arctic deviate significantly from the linear relationship. This supports previous studies demonstrating that regional precipitation responses are not simply linked to temperature through regional energy budget constraints, but also depend on other factors such as prevailing circulation patterns and remote teleconnections (Bollasina et al., 2014; Wilcox et al., 2018; Lewinschal et al., 2019). Overall, the above indicates the importance of changes in aerosol-related emissions in both global and regional precipitation changes. This is particularly true for Asia and Europe which represent the major sources of present-day aerosol-related emissions. In addition, aerosol changes are shown to have important influences on the ITCZ that tend to shift it towards the warmer hemisphere (Allen and Sherwood, 2011; Hwang et al., 2013; Allen and Ajoku, 2016; Acosta Navarro et al., 2017; Liu et al., 2018).

## 4. Discussion

### 4.1 Nonlinearities and the importance of background aerosol levels

Instead of linearly attributing the total aerosol changes into individual contributing factors, a "what-if" approach was adopted to develop the EDGAR retrospective emission scenarios (Crippa et al., 2016). This design is useful to assess the effectiveness of major drivers of emissions and allows us to show explicitly the policy-choice driven impacts, while accounting for nonlinear interplays between individual drivers. However, this approach adds extra nonlinearities to the results presented here in that, as discussed throughout this work, aerosol changes from energy use growth and technology advances do not add up to the total net 1970-2010 emission changes. This may suggest the existence of other factors taking effects, while it is almost impossible to attribute the residuals to such factors. Nevertheless, even when total emissions are linearly decomposed into individual contributing factors, it is reasonable to expect both the radiative forcing and climate responses to not linearly add up because of a variety of intertwined mechanisms. (e.g., the location-dependent lifetime of different aerosol species (Liu et al., 2012),

and the forcing efficacies (Kasoar et al., 2016; Aamaas et al., 2017)). At the global scale, despite the nonlinearities in aerosol/precursor emissions and AOD changes, the diagnosed global mean ERF and temperature responses roughly add up (Fig. 3). However, this is not the case for precipitation. When averaged regionally, the nonlinearities are more pronounced and can be seen through the pathway from emissions to AOD and ERF, and to temperature and precipitation responses.

In Sect. 3.1, we diagnosed the ERF associated with changes in each individual aerosol species as the differences between the baseline Fsst simulation (B10) and the ones where the targeted species (e.g., BC) are kept constant at their 1970 levels while the others are as prescribed in B10. We note that changes in both the spatial pattern and the global mean amount (Fig. S3) of the burden and AOD of the three aerosol species do not show appreciable differences to those in the experiment where all the three species change simultaneously (B10-B70). However, the ERF estimates do not linearly add up to the total. In fact, the

residual (0.14 W m$^{-2}$) is even larger in magnitude than the 1970-2010 total net aerosol ERF (-0.11 W m$^{-2}$). This reflects partly the nonlinear effect associated with the ratio of different aerosol species as well as the importance of background aerosol loadings. This is particularly important for BC whose effects depend also on the presence of sulphate and organic aerosols (Ramana et al., 2010). That is, given that aerosol species are internally-mixed in MAM3 (i.e. different chemical species are mixed within an aerosol particle), the hygroscopicity of aerosol particles is dominated by the volume of soluble species (organic

compounds and sulphate). This means that the nonlinearity in the isolated aerosol ERF may be a reflection of the aerosol scheme in CESM1. More specifically, BC particles tend to be coated with other species (e.g., sulphate, ammonium, and organic carbon) during ageing, thereby enhancing the absorption and the subsequent impacts on cloud microphysics, as well as amplifying their radiative forcing (Haywood and Ramaswamy, 1998; Kim et al., 2008; Chung et al., 2012; Wu et al., 2016). That is, the radiative forcing of BC may change with the ratio of BC to soluble aerosol species. Here, the ERF of BC is

diagnosed as the difference between the baseline experiment (B10) and that with BC held at the 1970 levels, leading the latter experiment to have a smaller ratio of BC to SO$_4$ and therefore smaller ERF. As a consequence, the ERF estimate due to the 1970-2010 changes in BC may be overestimated and may contribute to nonlinearities in the ERF of individual species. Note that these nonlinearities can be further enhanced by processes related to aerosol-cloud interactions, which are difficult to quantify (Fan et al., 2016; Forster et al., 2016).

Overall, the above discussion illustrates the importance of background aerosol concentrations and how models represent the mixing state in estimating the radiative forcing of aerosols. For example, we speculate that diagnosing the ERF of BC the other way round, namely, keeping all other aerosol species at 1970 levels while changing BC to 2010 levels, would likely result in different ERF estimates. Therefore, it is important to carefully bear in mind the model experiment set-up design when interpreting the ERF and climate responses associated with aerosol changes. For example, the single forcing experiments in

the Coupled Model Intercomparison Project (CMIP5; Taylor et al. (2012) ), the PDRMIP and other idealized aerosol perturbation experiments (Wang et al., 2015; Samset et al., 2016; Kasoar et al., 2018; Liu et al., 2018; Persad and Caldeira, 2018), as well as the upcoming AerChemMIP (Collins et al., 2017) model experiments all need to be interpreted in the context of their experiment designs.

**4.2 Caveats on the use of effective radiative forcing for aerosols**

The ERF is generally deemed to be a useful indicator of temperature changes (Shindell and Faluvegi, 2009; Myhre et al., 2013; Shindell et al., 2015; Forster et al., 2016; Lewinschal et al., 2019). Based on ERF, many metrics have been proposed to facilitate comparing the effectiveness of various forcing agents. Also, these metrics are appealing to quickly assess the climate outcomes of possible future emission pathways, and may hence provide useful information to policy-makers (Aamaas et al., 2017; Lewinschal et al., 2019). However, it is known that forcing and temperature response are not necessarily collocated, due to

many other climate processes and feedbacks such as the atmospheric and oceanic heat transport, and atmospheric circulation adjustments (Boer and Yu, 2003; Shindell et al., 2010; Bellouin et al., 2016; Persad and Caldeira, 2018). Specifically, ERF and the associated metrics may work for well-mixed forcing agents such as GHGs (Zhao et al., 2019b). However, they are misleading and may open to miss-interpretation when used for aerosols and some other short-lived climate forcers.

We stress here again that temperature responses do not necessarily follow the ERF of aerosols. In this work, the range (0.5-2.4 K $(W\,m^{-2})^{-1}$) of the global mean temperature response per unit ERF is even larger than that (0.1-1.4 K $(W\,m^{-2})^{-1}$) reported by Persad and Caldeira (2018). Also, our results suggest that the model simulated temperature response per unit aerosol ERF can differ considerably with even subtle differences in experiment design (e.g. with different amount of aerosols emitted in different locations at different timings). Further, due to the fact that aerosol schemes are represented differently across present generation climate models, it is highly likely that the sensitivities will differ across climate models. Therefore, as also pointed out by recent works (Persad and Caldeira, 2018; Lewinschal et al., 2019), the large divergence in the temperature response per unit ERF from aerosols highlights the need to use ERF and derivative metrics carefully for aerosols.

### 4.3 Implications for future climate projection

Reliable projections of future climate under different but equally plausible emission pathways are of utmost importance to better constrain the range of possible societal risks and response options. Unfortunately, there are still considerable challenges due to limitations and uncertainties in our understanding of many aspects of the Earth System (Knutti and Sedláček, 2013; Northrop and Chandler, 2014; Marotzke, 2019). Aerosols represent one of the largest sources of uncertainty (Boucher et al., 2013; Lee et al., 2016; Fletcher et al., 2018). Present-day anthropogenic aerosol-related emissions are largely influenced by sectors including power generation, industry and transport. However, in some of the future emission pathways, for example, the Tier-1 Shared Socioeconomic Pathways scenarios (SSP1; Gidden et al. (2018)), aerosol-related emissions are expected to decline drastically worldwide as we transit to non-fossil-fuel-based sources of energy, together with rapid implementation of air pollution control measures and new technologies. For example, mainly as a result of China's transition to a less energy-intensive society, for the first time the global coal consumption decreased in 2015 since the 1970s (World Energy Council, 2016). However, the timing and rate of such transitions are largely uncertain. On the other hand, it is also likely that aerosol-related emissions will increase, especially over some developing countries, under scenarios where high inequality exists between and within countries. For example, in SSP3, expanding industrial sectors over Southeast Asia may continue to rely on traditional energy sources such as coal for much of the 21$^{st}$ century. Also, it is possible that the world may continue to rely on fossil energy sources more strongly than expected over the coming years, given the concerns about nuclear energy after the Fukushima Daiichi nuclear disaster in March 2011. As a consequence, aerosol-related emissions from energy use in some regions may increase and therefore offset aerosol reductions elsewhere.

The above discussion reflects the large uncertainties (both spatially and temporally) in our understanding and estimates of future aerosol-related emission trajectories, given the possibility that very different future emission pathways may be adopted by different countries to compromise between climate/air pollution impacts and economic growth. Our findings may help better assess and interpret such uncertainties in future climate projections associated with changes in aerosols. First of all, the large impacts of present-day aerosol changes from the two competing drivers, as reported in this work, suggest that the major drivers (e.g., future energy structure and efficiency, air pollution control measurements, as well as technology progresses) of emission changes are likely to continue to play important roles in future climate projections. Secondly, uncertainties in future aerosol-related emission pathways combine with those of other climate forcing agents (e.g., greenhouse gas emissions and land-use changes). Such uncertainties influence the impacts of aerosol forcing through changing the background climate state (see Section 4.2; e.g., (Frey et al., 2017; Nordling et al., 2019; Stolpe et al., 2019)). More importantly, our results stress the importance of nonlinearities when comparing and assessing the impacts of different future aerosol-related emission pathways. This adds further caveats in interpreting future climate projections related to aerosol changes in addition to uncertainties in emission pathways of both aerosols and their precursors and GHGs.

## 5. Summary and conclusions

Using CESM1, time-slice simulations were carried out to investigate the ERF and climate impacts of 1970-2010 aerosol changes, focusing on two major policy-relevant emission drivers that compete: energy use growth and advances in emission control technology. The 1970-2010 anthropogenic aerosol changes generate a global mean ERF of $-0.11\pm0.14$ W m$^{-2}$. This is dominated by sulphate species, but the ERF estimates resolved into each individual species do not add up linearly to the total. The residual may be associated with the relative ratio of different aerosol species (Kim et al., 2008), as well as many other intertwined nonlinear processes linking aerosol changes to radiative forcing, and to temperature and precipitation responses. These nonlinearities highlight the importance that one must bear aerosol experiment designs carefully in mind when interpreting aerosol forcing and effects. In particular, the background concentration of both GHGs and aerosols may have strong influences on isolated aerosol effects using climate models (Regayre et al., 2018; Grandey and Wang, 2019).

1970-2010 energy use growth leads aerosols to increase over the Northern Hemisphere and Asia in particular, giving a global mean ERF of $-0.31\pm0.22$ W m$^{-2}$, and resulting in a global mean cooling ($-0.35\pm0.17$ K) and precipitation reduction ($-0.03\pm0.02$ mm day$^{-1}$). On the contrary, the avoided aerosol-related emissions due to technology advances generate a global mean ERF of $+0.21\pm0.23$W m$^{-2}$, and result in a global warming ($+0.10\pm0.13$ K) and precipitation increase ($+0.01\pm0.02$ mm day$^{-1}$). Change in aerosols and the resultant climate impacts are dominated by energy use growth over Asia but by technology advances over Europe and North America, while the global changes reflect competition between these two drivers. Compared to the rest of the world, temperature responses in the Arctic are noticeably amplified because of sea-ice and albedo related feedback processes (Navarro et al., 2016; Wobus et al., 2016; Dobricic et al., 2019). The large temperature responses are likely to be related to changes in aerosols over Europe and North America, while our results demonstrate that aerosol-related emissions from Asia may also play an important role (Westervelt et al., 2015; Wang et al., 2018; Dobricic et al., 2019). The temperature and precipitation responses to aerosol changes demonstrate influences of adjustments in atmospheric circulation induced by aerosol changes that can overwhelm the effects of local aerosol forcing. This is particularly important over higher latitudes such as the Arctic and Europe, and questions the usefulness of ERF as an indicator of the temperature response to aerosol forcing. We acknowledge the caveat/limitation of this study in that all our findings may be model dependent, which is particularly the case for aerosols, given the high degree of parameterisation and divergence in aerosol schemes across present generation climate models. We also note that CAM5 has a relatively larger aerosol forcing compared to other CMIP5 models (Allen and Ajoku, 2016; Malavelle et al., 2017; Toll et al., 2017; Zhou and Penner, 2017). These findings, therefore, need to be verified using other models, while identifying the possible underlying differences and reasons.

In conclusion, energy use growth and technology advances represent two major drivers of present-day aerosol changes, and have strong and competing impacts on present-day climate. We anticipate that there will be significant but uncertain changes in aerosol-related emissions over the coming decades driven by these two drivers. Also, there are a variety of nonlinearities in the effects of aerosols, originating from many factors including aerosol experiment design. All these uncertainties and nonlinearities may translate into even larger uncertainties in future climate projections and associated impacts. Given all the findings and implications laid out above, we strongly encourage model groups to better constrain the nonlinearities and uncertainties associated with aerosols in their climate models. Also, we encourage the wider research community to verify and further develop our findings in terms of aerosol-climate interactions and projections, as well as policy-relevant aerosol-related changes and their influences on air quality and associated socioeconomic impacts.

**Code and data availability**: This work uses the Community Earth System Model on the ARCHER UK National Supercomputing Service. The model outputs were pre-processed using netCDF Operator (NCO) and Climate Data Operator (CDO). The analysis is carried out using the Python programming language. Data presented here can be freely accessed (embargoed) through the Edinburgh DataShare (https://datashare.is.ed.ac.uk/handle/10283/3369).

**Author contributions:** D.S. conceptualized the project. M.C. provided the EDGAR emission scenarios. A.Z. M.B. and D.S. planned and designed the experiments. A.Z. carried out all model experiments, analysed the model outputs and produced all results. The manuscript was drafted by A.Z., and improved with inputs from all other co-authors.

**Competing interests:** The authors declare no conflict of interest.

**Acknowledgements:** D. S. Stevenson acknowledges support from the NERC grant NE/N003411/1 and NE/S009019/1. M. A. Bollasina was supported by the UK-China Research and Innovation Partnership Fund through the Met Office Climate Science for Service Partnership (CSSP) China as part of the Newton Fund (grant no. H5438500). This work used the ARCHER UK National Supercomputing Service (http://www.archer.ac.uk). The authors thank the Community Earth System Model project at NCAR. We are grateful to Gary Strand (NCAR) for providing the model dumps. We also thank the four anonymous reviewers for their very valuable comments.

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

**Table 1** Overview of the fully-coupled (Fcpd) and the paired simulation (Fsst) where sea surface temperature and sea ice are fixed. They are: the baseline 2010 (B10) simulation, fixing aerosol-related emissions in 1970 levels (SAA), stagnation of anthropogenic aerosol-related emissions from energy use in 1970 levels (SEN), and stagnation of aerosol-related emissions related to technology and abatement measures in 1970 levels (STC). All Fcpd simulations are run into equilibrium (numbers in brackets denote the lengths of model integrations in years), while all Fsst runs are integrated for 40 years. Only the last 30 years of each Fcpd/Fsst run are used for analysis. Note the difference in the integration lengths of Fcpd simulations, which is determined on the criterion that the top-of-the-atmosphere radiation imbalance no longer shows significant trends (stabilizing at around ~0.3 W m$^{-2}$ in this case) during the last few decades of each run (see the main text). The response to the best estimate of 1970-2010 anthropogenic aerosol-related emissions: Best Estimate = B10-SAA, energy use growth = B10–SEN; technology advances = B10–STC.

| Experiment (length of Fcpd/Fsst) | Greenhouse gases | Ozone | Natural aerosols | Anthropogenic aerosols |
|---|---|---|---|---|
| B10 (150/40) | 2010 | 2010 | 2010 | 2010 best estimate |
| SAA (120/40) | 2010 | 2010 | 2010 | 1970 best estimate |
| SEN (220/40) | 2010 | 2010 | 2010 | 2010 STAG_ENE |
| STC (170/40) | 2010 | 2010 | 2010 | 2010 STAG_TECH |

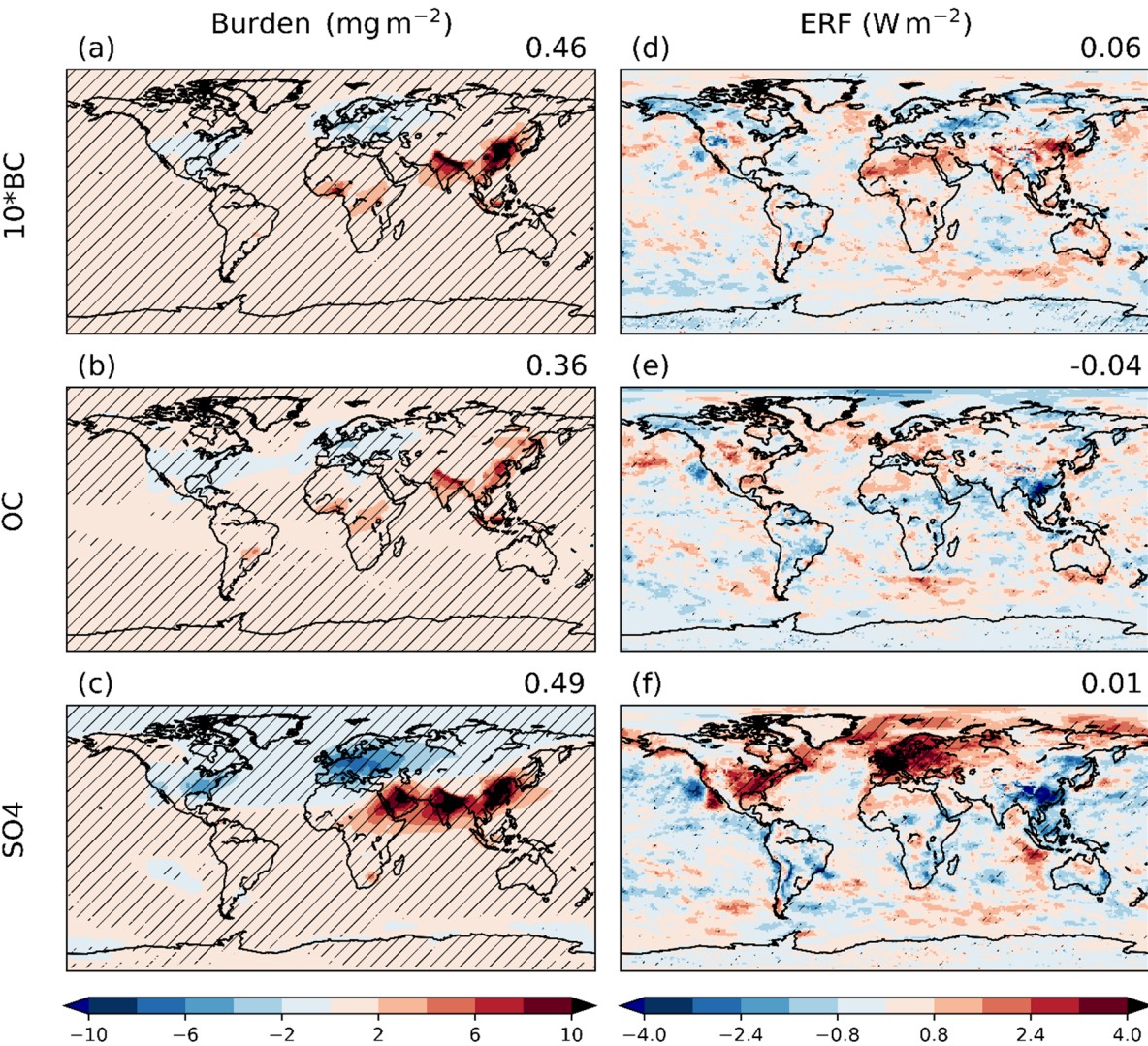

**Figure 1** Changes in aerosol burdens (mg m$^{-2}$, left) and the effective radiative forcing (ERF, W m$^{-2}$, right) associated with the best estimate of 1970-2010 changes in emissions of (**a, d**) black carbon (BC), (**b, e**) organic carbon (OC) and (**e, f**) sulphate species (SO$_4$). The numbers on the top right of each panel are the global means. NB the burden of BC (including the global mean value) is multiplied by a factor of 10 for legibility. The statistical significance at 5% level is calculated using the two-tailed student t-test and is denoted by the grey hatches.

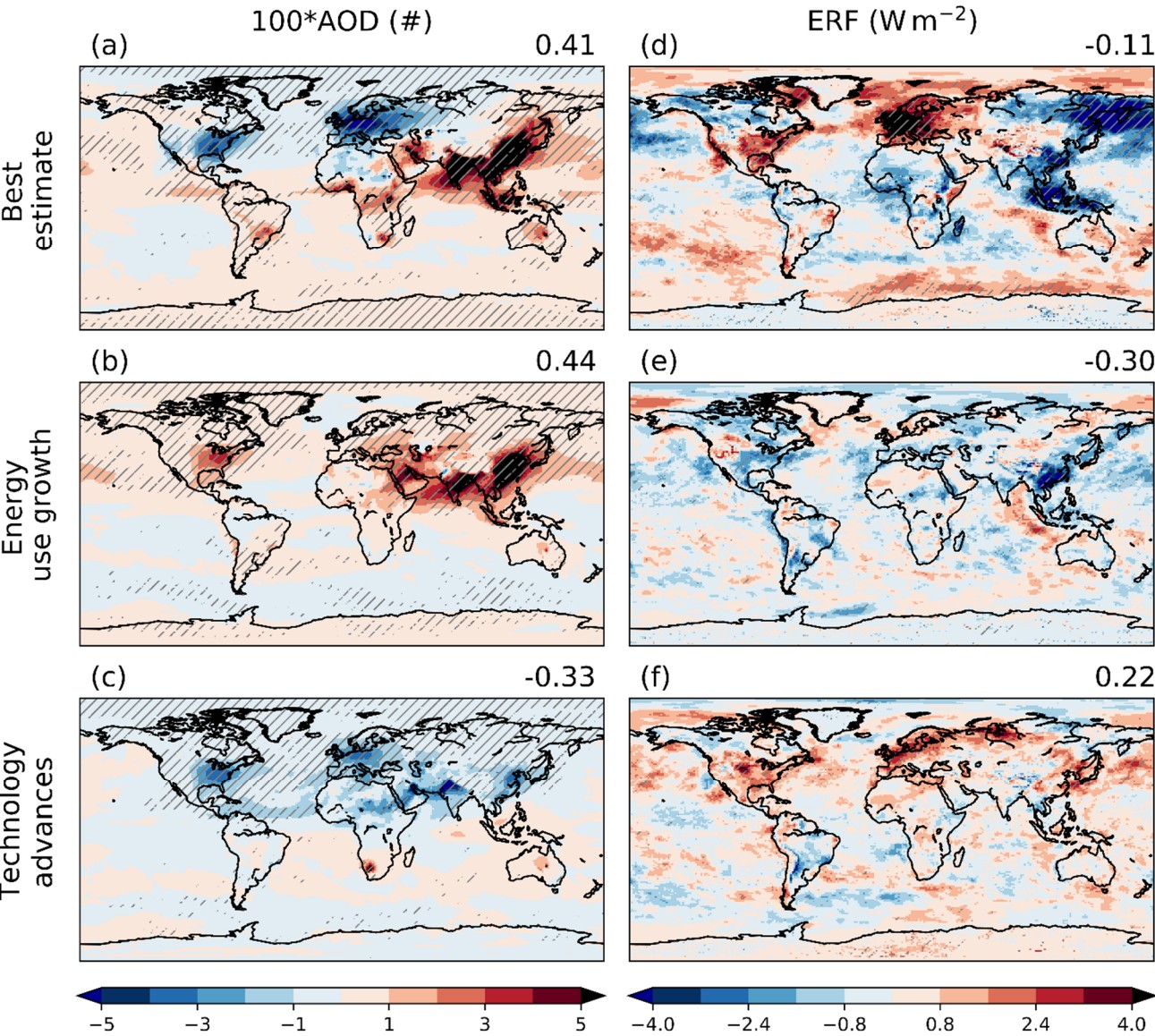

**Figure 2**. Changes in (**a-c**) 550-nm Aerosol Optical Depth multiplied by a factor of 100 (100*AOD) and (**d-f**) the effective radiative forcing (ERF, W m⁻²) in response to 1970-2010 anthropogenic aerosol emissions changes. They are: (**a, d**) the best estimate of total aerosol emission changes, (**b, e**) changes due to energy use growth and (**c, f**) changes due to advances in emission control technology. The numbers on the top right of each panel are the global means (NB again the AOD ones are multiplied by a factor of 100). The statistical significance at 5% level is calculated using the two-tailed student t-test and are denoted by the grey hatches

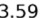

**Figure 3**. Area-weighted global and regional mean changes in (**a**) Aerosol Optical Depth (100*AOD), effective radiative forcing (ERF, W m⁻²), surface air temperature (SAT, K), and precipitation (Pr, mm day⁻¹). Error bars denote the 25ᵗʰ–75ᵗʰ percentile spread of the model uncertainty. Region definitions are as follows: Arctic (0°E-360°E, 60°N–0°N), Asia (65°E–145°E, 5°N-45°N), Europe (10°W–40°E, 30°N-70°N) and North America (190°E-300°E, 12°N-70°N). Colour conventions are: blue for responses to the best estimate of 1970-2010 anthropogenic aerosol emissions changes, red for responses to aerosol emission changes due to energy use growth and green for advances in emission control technology. NB carefully that the AOD values are multiplied by a factor of 100 for legibility.

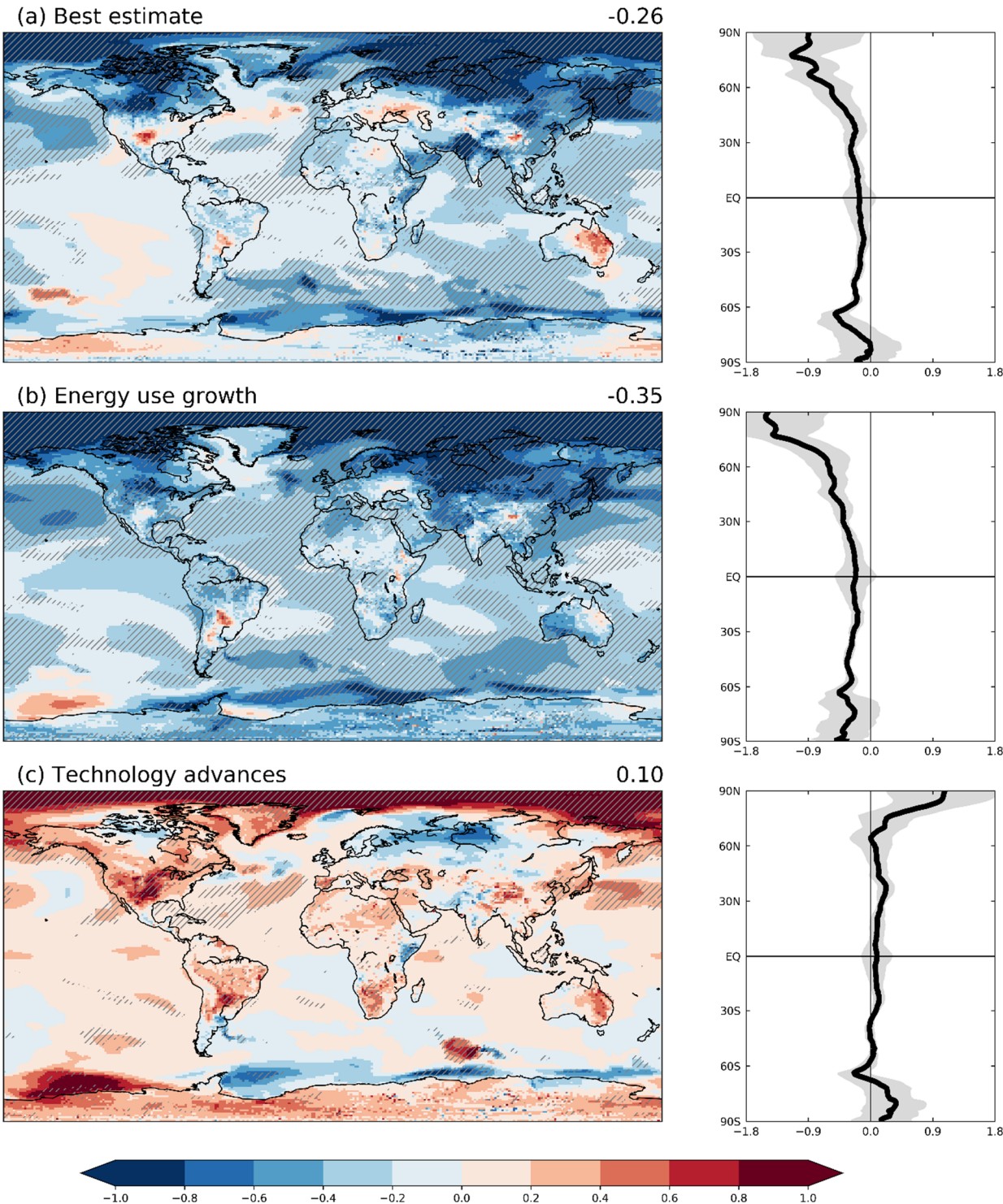

**Figure 4** Annual mean surface air temperature change (Δ K), in response to 1970-2010 anthropogenic aerosol emission changes. They are: (**a**) the best estimate of total aerosol emission changes, (**b**) changes due to energy use growth and (**c**) changes due to advances in emission control technology. The numbers on the top right of each panel are the global mean values. Also shown are the mean (solid) and standard deviation (30 model years; shadings) of the zonal mean temperature response. The statistical significance at 5% level is calculated using the two-tailed student t-test and is denoted by the grey hatches

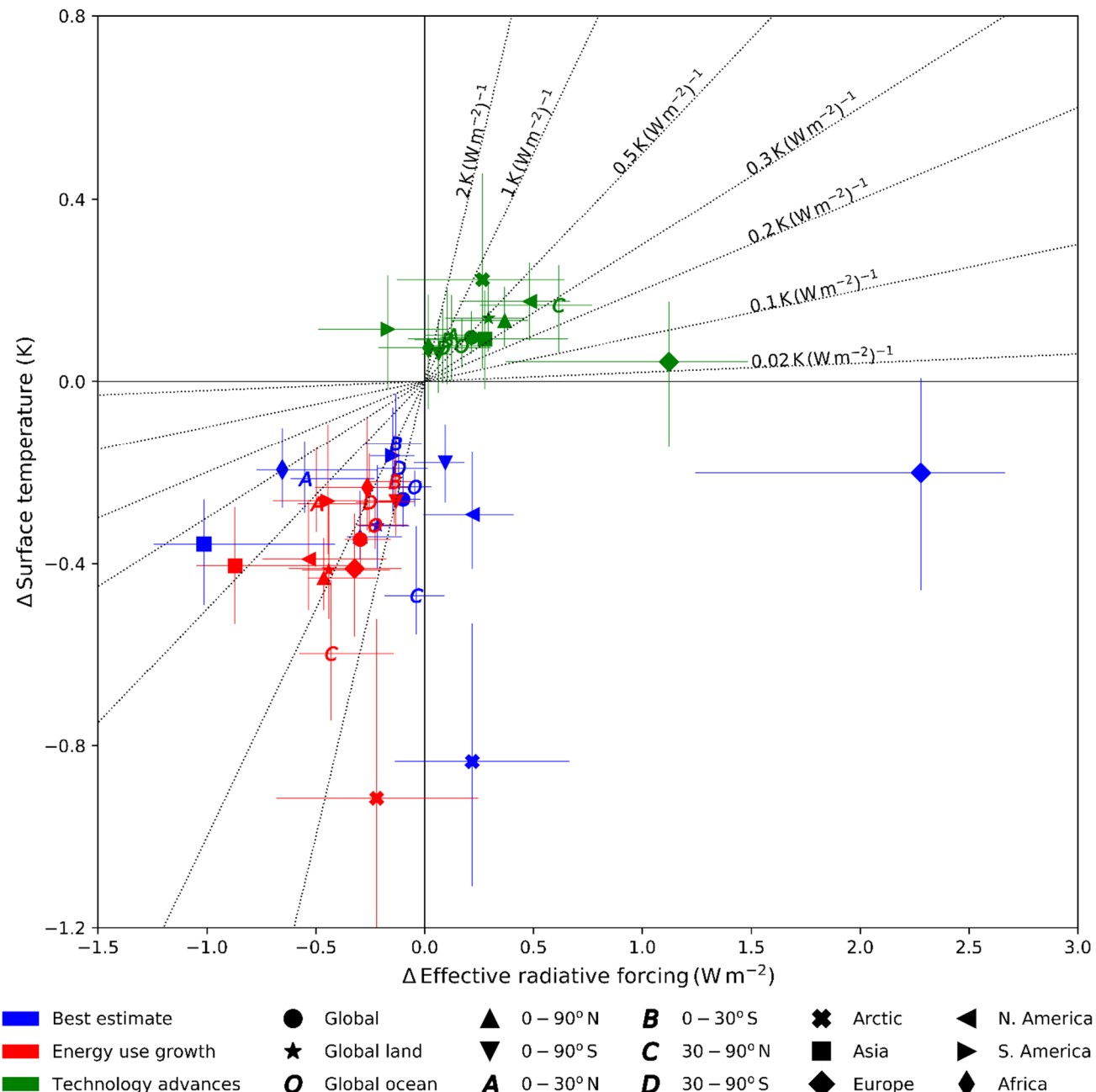

**Figure 5**. Scatterplot of surface air temperature responses ($\Delta$ K) vs. effective radiative forcing ($\Delta$ W m$^{-2}$). The error-bars represent the 25$^{th}$-75$^{th}$ percentile spread of the model uncertainty. The dashed slope lines crossing the origin indicate the sensitivities of the temperature response to ERF with a unit of K (W m$^{-2}$)$^{-1}$. Region definitions are as follows: Arctic (0°E-360°E, 60°N–0°N), Asia (65°E–145°E, 5°N-45°N), Europe (10°W–40°E, 30°N-70°N), North America (190°E-300°E, 12°N-70°N), South America (278°E-326°E, 56°S-12°N) and Africa (20°W-60°E,-35°S-25°N) plus latitudinal bands. Colour conventions are the same as Fig. 3.

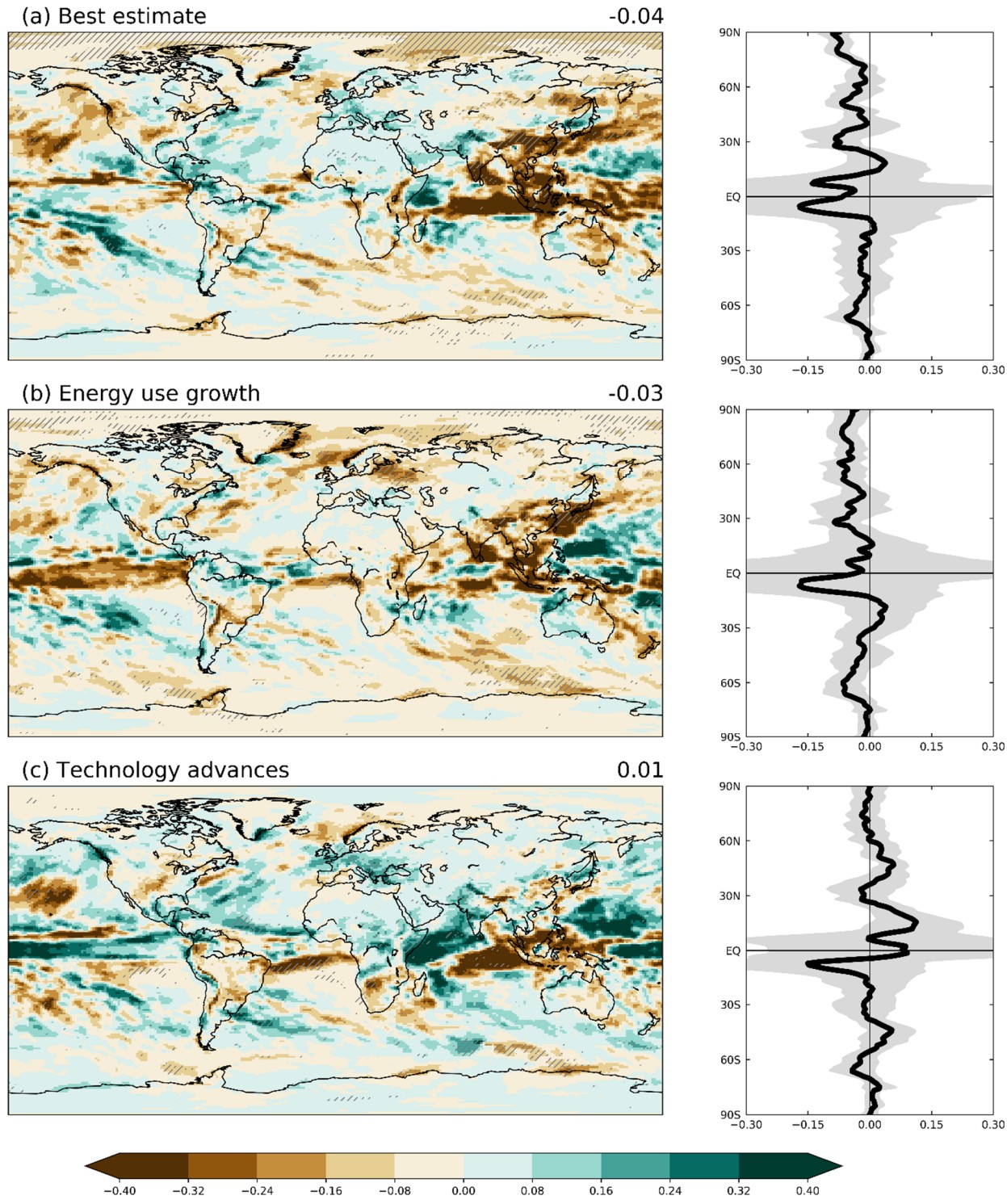

**Figure 6** The same as **Fig. 4**, but for precipitation response (Δ mm day⁻¹).