# Peer review of "Significant climate impacts of aerosol changes driven by growth in energy use and advances in emissions control technology"

_Atmospheric Chemistry and Physics, 2019_

## Referee Comment (RC1) · Anonymous Referee #1 · 14 Aug 2019

Summary

The authors conduct coupled ocean atmosphere and fixed SST aerosol simulations with the CESM, focused on 1970-2010 emission perturbations due to changes in energy growth and advances in emission controls. The latter is associated with a negative ERF, global cooling and reduced precipitation. The former is associated with the opposite responses. Regional impacts are also discussed, as is the non-linearity of the responses, and the limitations of using the ERF to diagnose aerosol temperature/precipitation responses.

Overall, the paper is well written, and the experiments and results are clearly presented. From my perspective, the result emphasized in the title of the manuscript is not that surprising (at least qualitatively). What appears to be most interesting is the significant amount of non-linearity in the responses. As discussed, this has important implications for experimental design to quantify aerosol climate responses, and also reinforces the difficulty in quantifying future aerosol-climate impacts.

Specific Comments

Figure 1. A bit confusing for BC. Is the BC ERF also increased by a factor of 10? I assume no (which is confirmed by the text), but the panel label shows "BCx10".

Section 2.2. Please include information on CESM's aerosol forcing, relative to other models. For example, CESM has a relatively large aerosol indirect effect. Which will subsequently impact these results. Allen and Ajoku calculated the aerosol effective radiative forcing (ERF; (W/mˆ2)) for 2000 relative to 1850, from CMIP5 models using the sstClim and the sstClimAerosol experiments. CESM yields -1.52 W/mˆ2, which is one of the largest aerosol ERFs. See Table 1 in "Future aerosol reductions and widening of the northern tropical belt" JGR, 2016.

Also, it is not acknowledged that these results are likely highly model dependent until the conclusions (L21 P11). I suggest making this point earlier in the manuscript (as well as in the conclusions).

P8 L15. See also "A 21st century northward tropical precipitation shift caused by future anthropogenic aerosol reductions" JGR 2015.

---

## Referee Comment (RC2) · Anonymous Referee #2 · 14 Aug 2019

The paper reports and discusses the results of a modeling effort to attribute aerosol climate impacts to changes in energy consumption as well as emission control technology. The window for analyses is 1970-2010 based on the EDGAR emission inventory. In the study, the authors designed two sets of modeling simulations using the Community Earth System Model or CESM. The first ser used prescribed sea surface temperature and sea ice for the purpose to derive the effective radiative effects or ERF of aerosols. The second set includes various equilibrium type of long integrations using coupled CESM with different aerosol configurations. Both methods are commonly used in climate studies. The result represents an interesting incremental progress by connecting the aerosol climate impact with sectional emissions of aerosol and aerosol

precursors. The paper itself is well organized despite certain presentation issues (see later comments). Its content is adequate for the reader of ACP. There are, however, still a few issues in the manuscript to be resolved before the paper can be accepted for publication.

A clear issue in presentation, mostly appears in Section 3, is missing the term of "change" throughout discussions from aerosol burden to ERF, and beyond. This confuses the difference between two time slices with the absolute quantity of aerosol burden and radiative effects. A few examples, "BC emissions generate a global mean positive radiative forcing of +0.06 W/m^2" (Pg. 5, LN 29); and "The global mean ERF of sulphate aerosols . . ." (Pg.6, LN 1). Regarding the precipitation change, it is not clear why the insignificant changes were highlighted even by quantity in, e.g., Abstract, while the much more profound regional changes were not mentioned there at all.

An interesting while somewhat puzzling result of this study is the nonlinearity revealed in several aspects related to aerosol, from ERF to model equilibrium sensitivity to aerosol ERF. The reason behind the fact that aerosol-species-based ERFs do not add up might have something to do with the (uniformly) internal mixing nature of the aerosol model where the hygroscopicity of aerosol is largely decided by organic carbon content due to its dominance in volume (ERF is largely a reflection of aerosol-cloud interaction or indirect effect of aerosols). Additional discussions are needed. Regarding the model's equilibrium sensitivity, could the different integration times in various simulations be at least a part of the reason responsible for the "nonlinearity"? Note that although the TOA forcing residual might be minimized throughout the quasi-equilibrium stage, ocean status such as SST evolution might still differ from time to time. Note also that aerosol forcing is rather small comparing to many internal factors of the model. Therefore, comparing simulations at different stages could likely introduce an arbitrary discrepancy in derived mean values. The authors should experiment using the same time slice for equilibrium analysis.

Minor comments. Pg. 2: LN 20, "forcing type", could be elaborated. LN 24, "ongoing

debates" on what topic? LN 28-29, "developing region", perhaps "developing countries" is better.

Pg. 4: Section 2.1, the relative changes of BC, OC, and sulphate from 1970 to 2010 referring to either 1970 or 2010 alongside relative ERF change should be provided. LN 23, "their number concentration", please note that for a modal aerosol model, the number concentration is defined for each mode rather than different aerosol compositions. LN 26, "to be coated", this could only be an assumption in performing certain calculation (e.g., optics) and is actually not necessarily consistent with the model's configuration.

Pg. 5, LN 16, the configuration of paired Fsst simulations should be listed in either Table 1 or a separate table.

Pg. 6, Section 3.2, according to Figure 4, it seems that majority of statistically significant changes in temperature appear over oceans rather than land.

Pg. 9, LN 2, "the residual (0.14 W/mˆ2)", could the authors elaborate on how to derive this residual?

Pg. 10, LN 1, "...temperature responses do not necessarily follow the ERF...", why? The scale of temperature response to ERF or the "equilibrium sensitivity" could differ from case to case, but for the same forcing agent in the same model, it should be the same, in other words, the temperature response should be always proportional to (or follow) ERF.

Pg. 10, LN 18-22, "...it is also likely that aerosol emissions will increase...", this discussion actually raises an interesting issue that recent increase of aerosol emissions could occur not only in developing but also developed countries. Note that the EDGAR estimate used here is up to 2010, a year before Fukushima Daiichi nuclear disaster in March 2011. Due to the closure of nuclear facilities from Europe to East Asia following that event, it is likely that in recent years coal burning has already come back in

many of these regions because of obvious shortage in energy supply otherwise from renewables alone.

---

## Referee Comment (RC3) · Anonymous Referee #3 · 16 Aug 2019

The study uses the CESM global climate model to quantify the impact on effective radiative forcing, surface temperature, and precipitation rates of increases in aerosol emissions due to energy generation and decreases in emissions due to technological advances over the period 1970-2010. The authors find that technological advances only partly offset impacts of energy generation. They also highlight the non-linearity of effective radiative forcing and diversity in temperature sensitivities.

The paper is well written, and the figures support the discussion well. Figure 5 is perhaps the most novel aspect of the study and acts as a very efficient summary of the main findings. There are however results I do not understand, and the discussion

could be improved in places. For these reasons, I recommend minor revisions.

**1  Main comment**

- I understand why temperature sensitivities, as defined in the study, could be negative (i.e. located in the top left corner of Figure 5) locally because (i) albedo and circulation feedbacks mean that temperature responses are not collocated with ERF and (ii) both ERF and surface temperature distributions are inherently noisy. But I cannot understand how temperature sensitivities can be negative globally, as stated for the best estimate (Page 7 line 16). Is that really the case? According to Fig 3 and 4, ERF is +0.1 W m$^{-2}$ and DeltaT is +0.09, so the sensitivity should be positive. If there are negative global sensitivities, I would be concerned that the model does not conserve energy. . .

**2  Other comments**

- Abstract, page 1, line 19: " must be interpreted in the context of experiment designs" is a cryptic statement that is in some way true of all experiments. I would suggest rewriting, perhaps to something like "must be interpreted in the context of the reference baseline".

- Page 3, lines 3–4: That statement could be toned down – improving air quality is getting harder, and climate feedbacks (e.g. this year's record Arctic wildfires) may negate parts of the decreases.

- Page 3, line 10: The reason why the impact of future changes in aerosols have not been looked in isolation before is that (i) uncertainties in emission pathways

also affect other forcings, so aerosols are not alone in that respect, and (ii) considering all forcings together makes sense because sectors emit a cocktail of gaseous and particulate species. So the authors should make a better case for what we really gain at considering aerosols in isolation.

- Page 3, lines 19–21: What did the Turnock study find?

- Section 2.2: Which aerosol radiative forcing mechanisms does the model represent?

- Page 5, line 5 : permutations -> perturbations

- Page 5, line 29: Technically speaking, a decrease of $-4$ is an increase!

- Captions of Fig 2 and 3: "raised up" is unnecessarily ambiguous. Simply say "multiplied"

- Page 6, line 15 : Why is there an increase over Southern Africa? It looks statistically significant. Are there technological advances that increase emissions?

- Page 6, lines 22–24: That statement is not obvious to me. The area of strongest cooling, in Siberia, looks to be to the north of where emissions changes are probably located, in China. (It would really help to see maps of emission changes.)

- Page 6, line 31: Is the enhancement again due to sea-ice albedo?

- Caption of Fig 5: "error bars represent model uncertainty" – how is that calculated?

- Section 4.1: An alternative conclusion of that section is that trying to decompose ERFs and response into component-based estimates is misleading in an internal mixture context. Doing so is just not as informative as it first appears. Source-based decomposition would be more useful and, perhaps, less subject to non-linearities.

- Page 11, line 11: "Despite the global mean negative forcing from 1970-2010 aerosol changes" – but according to Fig 3, your best estimate ERF is positive.

- Page 11, line 21–24: Beyond model dependence, being critical of one's model is also useful. CAM is not the best-behaved model in Malavelle et al. 2017 doi:10.1038/nature22974, strongly overestimating rapid adjustments to aerosol-cloud interactions. But then even the best-behaved model gets the wrong rapid adjustments compared to observations Toll et al. 2017 doi: 10.1002/2017GL075280. That would imply that ERFs, and the subsequent temperature and precipitation responses, are too strong in that model.

---

## Referee Comment (RC4) · Anonymous Referee #4 · 17 Aug 2019

This paper assesses the influence on radiative forcing, temperature and precipitation of changes in anthropogenic aerosol emissions between 1970 and 2010 in the CESM1-CAM5 model. The paper decomposes the effects into the contribution from aerosol emissions increases due to increased energy use and the contribution form aerosol emissions decreases due to technological advances and air quality abatement initiatives over the same time period. The authors find that the two effects have partially cancelled each other over the analyzed period and do so nonlinearly and non-uniformly.

The paper is well written overall. There are several useful and interesting points made. The experiment design is clean and well thought out. However, I find that the motivation and certain aspects of the analysis require additional depth, in order to make the novelty of the paper clearer. I am also concerned that certain aspects of the results seem counterintuitive, suggesting problems with the model set-up (see Other Major Comments 2 and 3 below). At present it is not entirely clear what new the authors are adding to the conversation, though I do believe that this analysis has the potential to be a useful contribution with some additional depth. I believe that major revisions to provide further depth would be needed for the paper to merit publication in ACP. These are discussed further below:

- The best estimate results shown in this manuscript are essentially an assessment of impact of aerosols on climate over the last 40 years, which has been conducted extensively in the existing literature. The main novelty in the paper, therefore, lies in the decomposition into the two factors described above. However, I believe the authors must do more to center and frame this decomposition.

- At present, the authors do not make a strong case for why decomposing the total effect into these two components is a valuable exercise. This should be explicitly laid out in the introduction.

- The authors need to be clearer that the assessment of last 20th century effects of aerosols has been widely done elsewhere and provide this more as background, acknowledging where their work is replicative of and consistent with that done elsewhere, and focus more thoroughly on unique insight gained through the decomposition.

- At the same time, the analysis of the decomposition should be further deepened. Although the authors report many signals, they do not provide any depth of analysis on why these signals emerge. Given how many other studies have analyzed the role of aerosol increases and decreases in the second half of the 20th century and made many of the same points the authors make here, simply reporting

these signals is not sufficient. Where do the two effects counteract or reinforce each other in ERF, temperature, or precipitation? Why does this occur? What are the implications for future anticipated shifts in aerosol emissions due to the two effects?

- A main result that the authors emphasize is the result that the individual ERFs of the aerosol species do not add up linearly, giving rise to the authors' repeated statement that ERF calculations should be considered to be dependent on experiment set-up. While this is in some ways a fairly obvious statement, it would be useful to have this explored in further detail. However, the authors do not provide any explanation for this result. I believe the authors must assess the mechanism behind this more thoroughly for it to be a valuable contribution. Is the nonlinearity of the ERF when the aerosols are emitted separately versus together a result of changes in the total atmospheric burden or of changes in the spatial distribution and resulting radiative interactions? The authors allude briefly to the total burdens of each species being "identical" in the separate and combined simulations (P8, L20), but do not show it. This should be explicitly shown. The remaining explanations that are currently only suggested in Section 4.1 should be explicitly assessed.

**Other Major Comments:**

1. Confidence intervals should be provided for all global-mean values.

2. I agree with Anonymous Referee 3 that the negative global mean temperature sensitivity value in the best estimate case seems to violate basic energy conservation. How can a positive global-mean ERF result in a negative global-mean temperature change or vice versa? This suggest a major issue with the model formulation. If it is somehow robust, it needs to be explained.

3. Figure 1: Why does the large increase in SO4 burden result in a positive ERF? This needs to be explained, particularly when compared to the larger negative ERF of the smaller (and less reflective) OC burden.

4. Section 4.3 is largely a literature review on the formulation of aerosol emissions scenarios and does not explore what implications the results of this analysis have for future projections. How may their findings "help better assess and interpret such uncertainties in future climate projections"?

**Minor Comments:**

- The authors use a somewhat confusing formulation throughout the paper in which they refer to both the aerosol increases from increased energy use and the aerosol decreases from technological advances as "aerosol emissions" (e.g. P8, L23; P10, L26). I recommend rewording these to something like "aerosol emissions changes" throughout, to make clearer that it is in some case the absence rather than the presence of aerosol emissions that is being analyzed.

- The model and experiment design section would benefit from a clearer description in the text of what the different model simulations are and what scenarios they are intended to test. This is provided at the moment entirely in the caption to Table 1, but should be at least summarized in Section 2.2.

- It is not entirely clear what the residual emissions are that are occurring in the B10 simulation but that are not captured in the SEN and STC simulations. It would help for this to be more thoroughly described and for its magnitude relative to the emissions changes to be provided, especially given the authors repeated references to it.

- P3, L3-4: recommend replacing "will" with "may", as certain of the Shared Socioeconomic Pathways do simulate regional increases in anthropogenic aerosol

emissions.

- P5, L5: "where the climate system equilibrates to imposed permutations but the deep ocean" ← this phrase is unclear

- P7, L30: The drying over Europe as well as Asia seems counterintuitive, since all the other analysis suggested opposite trends (in burden, ERF, and temperature) over Europe and Asia. This bears further explanation.

- P8, L7-9: I'm fairly certain the estimate from this paper and from PDRMIP have almost entirely overlapping error bounds, making this difference hardly worth noting. I would rather read this as suggesting that the precipitation sensitivity is very well aligned with that found by PDRMIP. This is one reason why the authors need to provide confidence intervals on their global-mean values.

- P8, L10-12: "This may suggest that regional..." This statement seems deeply obvious and well-established in the community. Recommend cutting.

- Figure 1: It is not clear that the presence of hatching means that it is statistically significant rather than no statistically significant until one reads the text. Recommend rewording L5-6 to be clearer. It is also quick difficult to see the statistical significance hatching over the ERF values – recommend a different color (e.g. light grey) for the hatch lines.

- Figure 5: It is incredibly difficult to visually parse the symbols. The Global and Global Ocean symbols are essentially indistinguishable in the crowded areas of the graph. Very few of these are actually discussed in the manuscript, and I recommend removing some of the symbols (perhaps the latitude band symbols) to make this figure legible.

**Typographic comments:**

- There are several instances of in-line citations still being enclosed by parentheses that should be corrected (P4: L27, L28, L31)

- P7, L13: "domain" → "domains"

- P9, L12-13: "is difficult to be quantified" → "are difficult to quantify"

- The paper contains a couple stub sentences without verbs: p4, L14-15 and p8, L27-28

---

## Author Comment (AC1) · 25 Oct 2019

We are grateful to the reviewer for their interest and comments on the paper. These comments are very valuable and have helped improve the manuscript. Here we outline how we have addressed these comments in the revised manuscript. The newly added discussions and rephrased sentences have been highlighted in green in our replies below.

The authors conduct coupled ocean atmosphere and fixed SST aerosol simulations with the CESM, focused on 1970-2010 emission perturbations due to changes in en-ergy growth and advances in emission controls. The latter is associated with a neg-ative ERF, global cooling and reduced precipitation. The former is associated withthe opposite responses. Regional impacts are also discussed, as is the non-linearity of the responses, and the limitations of using the ERF to diagnose aerosol tempera-ture/precipitation responses.

Overall, the paper is well written, and the experiments and results are clearly presented. From my perspective, the result emphasized in the title of the manuscript isnot that surprising (at least qualitatively). What appears to be most interesting is the significant amount of non-linearity in the responses. As discussed, this has important implications for experimental design to quantify aerosol climate responses, and also reinforces the difficulty in quantifying future aerosol-climate impacts.

Specific Comments

1.      Figure 1. A bit confusing for BC. Is the BC ERF also increased by a factor of 10? I assume no (which is confirmed by the text), but the panel label shows "BCx10".

We thank the reviewer for pointing this out. Indeed, BC ERF is not multiplied by 10. We have modified the Figure with the label changed to "BC".

2.      Section 2.2. Please include information on CESM's aerosol forcing, relative to other models. For example, CESM has a relatively large aerosol indirect effect. Which will subsequently impact these results. Allen and Ajoku calculated the aerosol effective radiative forcing (ERF; (W/m^2)) for 2000 relative to 1850, from CMIP5 models using the sstClim and the sstClimAerosol experiments. CESM yields -1.52 W/m^2, which is one of the largest aerosol ERFs. See Table 1 in "Future aerosol reductions and widening of the northern tropical belt" JGR, 2016.

In combination with comment #3, we add the following sentences following P4 L26: "Note that CESM1 (CAM5) has a relatively larger aerosol forcing than other CMIP5 models, likely due to the large cloud adjustments through cloud water path in MAM3 (Allen and Ajoku, 2016; Malavelle et al., 2017; Zhou and Penner, 2017). In light of this and considering the overall uncertainties in the representation of aerosol effects, we underscore that all results and discussions below should be interpreted in the context of CESM1-CAM5."

3.      Also, it is not acknowledged that these results are likely highly model dependent until the conclusions (L21 P11). I suggest making this point earlier in the manuscript (as well as in the conclusions).

Please refer our reply to comment #2

4.      P8 L15. See also "A 21st century northward tropical precipitation shift caused by future anthropogenic aerosol reductions" JGR 2015.

We thank the reviewer for pointing us to Allen (2015) which is cited in the revised manuscript.

**References**

Allen, R. J. (2015), A 21st century northward tropical precipitation shift caused by future anthropogenic aerosol reductions, Journal of Geophysical Research: Atmospheres, 120(18), 9087-9102.
Allen, R. J., and O. Ajoku (2016), Future aerosol reductions and widening of the northern tropical belt, Journal of Geophysical Research: Atmospheres, 121(12), 6765-6786.
Malavelle, F. F., J. M. Haywood, A. Jones, A. Gettelman, L. Clarisse, S. Bauduin, R. P. Allan, I. H. H. Karset, J. E. Kristjánsson, and L. Oreopoulos (2017), Strong constraints on aerosol–cloud interactions from volcanic eruptions, Nature, 546(7659), 485.
Zhou, C., and J. E. Penner (2017), Why do general circulation models overestimate the aerosol cloud lifetime effect? A case study comparing CAM5 and a CRM, Atmospheric Chemistry and Physics, 17(1), 21-29.

---

## Author Comment (AC2) · 25 Oct 2019

We are grateful to the reviewer for their interest and comments on the paper. These comments are very valuable and have helped improve the manuscript. Here we outline how we have addressed these comments in the revised manuscript. The newly added discussions and rephrased sentences have been highlighted in green in our replies below.

The paper reports and discusses the results of a modelling effort to attribute aerosol climate impacts to changes in energy consumption as well as emission control technology. The window for analyses is 1970-2010 based on the EDGAR emission inventory. In the study, the authors designed two sets of modelling simulations using the Community Earth System Model or CESM. The first set used prescribed sea surface temperature and sea ice for the purpose to derive the effective radiative effects or ERF of aerosols. The second set includes various equilibrium type of long integrations using coupled CESM with different aerosol configurations. Both methods are commonly used in climate studies. The result represents an interesting incremental progress by connecting the aerosol climate impact with sectional emissions of aerosol and aerosol precursors. The paper itself is well organized despite certain presentation issues (see later comments). Its content is adequate for the reader of ACP. There are, however, still a few issues in the manuscript to be resolved before the paper can be accepted for publication.

While revising the manuscript, we realised that the isolation of the effects of aerosol changes in the "best estimate" experiment was probably biased due somehow to the experimental set-up. To address this issue, we have carried out a new experiment, which is now used throughout the revised manuscript (including Figures and texts). Note that, while this has resulted in different estimates of the impacts of aerosol changes in the best estimate case, it does not have any bearing on the major findings of this study on the comparison of the two different retrospective emission scenarios.

1. A Clear issue in presentation, mostly appears in Section 3, is missing the term of "change" throughout discussions from aerosol burden to ERF, and beyond. This confuses the difference between two time slices with the absolute quantity of aerosol burden and radiative effects. A few examples, "BC emissions generate a global mean positive radiative forcing of +0.06 W/m^2" (Pg. 5, LN 29); and "The global mean ERF of sulphate aerosols…" (Pg.6, LN 1). Regarding the precipitation change, it is not clear why the insignificant changes were highlighted even by quantity in, e.g., Abstract, while the much more profound regional changes were not mentioned there at all.

We thank the reviewer for pointing these issues out. We have revised the entire manuscript accordingly.

Regarding the abstract, we feel it is important to mention the global mean changes in both temperature and precipitation. Yet, in response to the reviewer's comment, we added a sentence in the Abstract following L15 as "Despite the relatively small changes in global mean precipitation, these two emission drivers have profound impacts at regional scales, in particular over Asia and Europe".

2.       An interesting while somewhat puzzling result of this study is the nonlinearity revealed in several aspects related to aerosol, from ERF to model equilibrium sensitivity to aerosol ERF. The reason behind the fact that aerosol-species-based ERFs do not add up might have something to do with the (uniformly) internal mixing nature of the aerosol model where the hygroscopicity of aerosol is largely decided by organic carbon content due to its dominance in volume (ERF is largely a reflection of aerosol-cloud interaction or indirect effect of aerosols). Additional discussions are needed. Regarding the model's equilibrium sensitivity, could the different integration times in various simulations be at least a part of the reason responsible for the "nonlinearity"? Note that although the TOA forcing residual might be minimized throughout the quasi-equilibrium stage, ocean status such as SST evolution might still differ from time to time. Note also that aerosol forcing is rather small comparing to many internal factors of the model. Therefore, comparing simulations at different stages could likely introduce an arbitrary discrepancy in derived mean values. The authors should experiment using the same time slice for equilibrium analysis.

We agree with the reviewer that the nonlinearity in ERFs may be related to the internal mixing state of the different aerosol species. We also agree with the reviewer regarding the potential role of different oceanic states on the inferred equilibrium of the various experiments.

In response to the comment, we expand P9 Ls3-7 into "This reflects partly the nonlinear effect associated with the mixing state of different aerosol species as well as the importance of background aerosol loadings. This is particularly important for BC whose effects depend also on the presence of sulphate and organic aerosols (Ramana et al., 2010). That is, given that aerosol species are internally-mixed in MAM3 (i.e. different chemical species are mixed within an aerosol particle), the hygroscopicity of aerosol particles is dominated by the volume of soluble species (organic compounds and sulphate). This means that the nonlinearity in the isolated aerosol ERF may be a reflection of the aerosol scheme in CESM1. More specifically, BC particles tend to be coated with other species during ageing, thereby enhancing the absorption effects and the subsequent impacts on cloud microphysics, as well as amplifying their radiative forcing (Haywood and Ramaswamy, 1998; Kim et al., 2008; Chung et al., 2012; Wu et al., 2016). …".

Regarding the use of different simulation periods in the analysis, we feel necessary to provide further motivations for this choice. As stated at P5 Ls 7-10, each fully-coupled model simulation was considered in equilibrium when the TOA net radiation was devoid of any significant trends (less than 5% relative to the mean values stabilizing at ~0.3 W m$^{-2}$) during a 30-50 period following previous works (*Samset et al.*, 2016; *Samset et al.*, 2018). This period was then used in the analysis. Note that our simulations are longer (~150 vs. 100 years) and with a smaller TOA radiation imbalance (~0.3 vs. above 0.5 W m$^{-2}$) compared to those analysed

in *Samset et al.* (2018), meaning that our integrations are even closer to equilibrium. While it is certainly possible to analyse the same time period in all experiments, we argue that this would lead to examining potentially different equilibrium states (as defined above), and thus to potential misinterpretations of the differences among them. Therefore, we believe it is more appropriate to compare the same equilibrium states rather than the same time period. In fact, this reflects a very important issue regarding the time scale of different forcing agents in climate models, a topic which deserves new and deeper analysis in the future.

Minor comments.

3.      Pg. 2: LN 20, "forcing type", could be elaborated. LN 24, "ongoing debates" on what topic? LN 28-29, "developing region", perhaps "developing countries" is better.

In L20, by "Forcing type" we mean different aerosol species (i.e., BC, $SO_4$ and organic compounds). To make this clearer, we reworded "forcing type" into "aerosol species".

In L24, the "ongoing debates" refers to the sentences in L25-27. Namely, ongoing debates on whether aerosol forcing has larger impacts on mean climate and climate extremes compared to GHG. We have now expanded the sentence "despite ongoing debates" into "despite ongoing debates as to whether aerosol forcing has larger impacts on mean climate and climate extremes compared to GHG.".

"Developing region" has been reworded into "developing countries".

4.      Pg. 4: Section 2.1, the relative changes of BC, OC, and sulphate from 1970 to 2010 referring to either 1970 or 2010 alongside relative ERF change should be provided. LN 23, "their number concentration", please note that for a modal aerosol model, the number concentration is defined for each mode rather than different aerosol compositions. LN 26, "to be coated", this could only be an assumption in performing certain calculation (e.g., optics) and is actually not necessarily consistent with the model's configuration.

We thank the reviewer for all the suggestions.

Since the 1970-2010 changes in BC, OC and $SO_2$ emissions have been thoroughly presented in a published paper by one of our co-authors (*Crippa et al.* (2016)), we have added Figure S1 in the supplement to show emission changes, and decided to direct the readers to *Crippa et al.* (2016) by for more details. As such, we added the sentence " For the 1970-2010 changes in emissions of each individual aerosol/precursor species, please refer to Figure S1 in the supplementary file and *Crippa et al.* (2016). " Following P4 L7,

P4 L 23: "Several aerosol species (sulphate, organic carbon (OC), black carbon (BC), sea-salt, and dust) are simulated and their number concentrations and mass are prognostically calculated" has been slightly rephrased into "Several aerosol species (sulphate, organic carbon

(OC), black carbon (BC), sea-salt, and dust) are simulated, and their number concentration and mass are prognostically calculated for each aerosol mode".

To avoid confusion, we deleted the statement on BC at P4 L 25-26.

5.     Pg. 5, LN 16, the configuration of paired Fsst simulations should be listed in either Table 1 or a separate table.

We thank the reviewer for the suggestions. We have updated Table 1 to include the Fsst runs. Also note as mentioned above, we have modified the table to account for the new experiment used to isolate the aerosol effects for the best-estimate case scenario.

**Table 1** Overview of the fully-coupled (Fcpd) and the paired stimulation (Fsst) where sea surface temperature and sea ice are fixed. They are: the baseline 2010 (B10) simulation, fixing aerosol-related emissions in 1970 levels (SAA), stagnation of anthropogenic aerosol-related emissions from energy use in 1970 levels (SEN), and stagnation of aerosol-related emissions related to technology and abatement measures in 1970 levels (STC). All Fcpd simulations are run into equilibrium (numbers in brackets denote the lengths of model integrations in years), while all Fsst runs are integrated for 40 years. Only the last 30 years of each Fcpd and Fsst run are used for analysis. Note the difference in the integration lengths of Fcpd simulations, which is determined on the criterion that the top-of-the-atmosphere radiation imbalance no longer shows significant trends (stabilizing at around ~0.3 W m$^{-2}$ in this case) during the last few decades of each run (see the main text). The response to the best estimate of 1970-2010 anthropogenic aerosol-related emissions: best estimate = B10-SAA. Similarly, energy use growth = B10–SEN; technology advances = B10–STC.

| Experiment (length of Fcpd/Fsst) | Greenhouse gases | Ozone | Natural aerosols | Anthropogenic aerosols |
|---|---|---|---|---|
| B10 (150/40) | 2010 | 2010 | 2010 | 2010 best estimate |
| SAA (120/40) | 2010 | 2010 | 2010 | 1970 best estimate |
| SEN (220/40) | 2010 | 2010 | 2010 | 2010 STAG_ENE |
| STC (170/40) | 2010 | 2010 | 2010 | 2010 STAG_TECH |

6.     Pg. 6, Section 3.2, according to Figure 4, it seems that majority of statistically significant changes in temperature appear over oceans rather than land.

The reviewer is correct in pointing out that the majority of statistically significant temperature changes are over the oceans rather than land. This is especially the case for the energy use experiment, and may be explained by the fact that the equilibrium climate responses are mainly related to the ocean.

We add a sentence to comment on this following P6 L20 as "It can be seen that the majority of statistically significant temperature changes in response to aerosol changes are over the ocean rather than the land. This is particularly true for the energy use experiment, and may reflect the fact that the equilibrium climate response is dominated by the slow response of the ocean."

7.     Pg. 9, LN 2, "the residual (0.14 W/m^2)", could the authors elaborate on how to derive this residual?

First of all, we apologise that the number on the top right of Figure 2d was incorrectly reported as 0.10 W m$^{-2}$. The correct value is -0.11 W m$^{-2}$ (see Figure 3b). We feel this may have confused the reviewer, and have corrected this in the revised version.

The ERF due to changes in BC (0.06 W m$^{-2}$), OC (-0.04 w m$^{-2}$), and SO$_4$ (0.01 W m$^{-2}$) add up to 0.03 W m$^{-2}$. The difference between the sum of individual aerosol ERFs in Figure 1 (0.03 W m$^{-2}$) and the total in Figure 2b (-0.11 W m$^{-2}$) therefore produces a residual of -0.14 W m$^{-2}$.

8.      Pg. 10, LN 1 temperature responses do not necessarily follow the ERF...", why? The scale of temperature response to ERF or the "equilibrium sensitivity" could differ from case to case, but for the same forcing agent in the same model, it should be the same, in other words, the temperature response should be always proportional to (or follow) ERF.

We understand the reviewer's point. However, as also demonstrated by other recent works (Persad and Caldeira, 2018; Lewinschal et al., 2019), the link between ERF and temperature response may hold particularly for long-lived and spatially homogeneous forcing factors, such as greenhouse gases, but may be weaker for short-lived aerosols which are, by nature, highly heterogeneous and involve complex atmospheric circulation adjustments (*Shindell and Faluvegi*, 2009; *Shindell et al.*, 2010).

9.      Pg. 10, LN 18-22, "...it is also likely that aerosol emissions will increase...", this discussion actually raises an interesting issue that recent increase of aerosol emissions could occur not only in developing but also developed countries. Note that the EDGAR estimate used here is up to 2010, a year before Fukushima Daiichi nuclear disaster in March 2011. Due to the closure of nuclear facilities from Europe to East Asia following that event, it is likely that in recent years coal burning has already come back in many of these regions because of obvious shortage in energy supply otherwise from renewables alone.

We thank the reviewer for providing very interesting insights.

We expand L18-23 into "On the other hand, it is also likely that aerosol emissions will increase, especially over some developing regions, under scenarios where high inequality exists between and within countries. For example, in SSP3, expanding industrial sectors over Southeast Asia may continue to rely on fossil energy sources such as coal. Also, it is possible that the world may continue to rely on fossil energy sources more strongly than expected over the coming years, given the concerns about nuclear energy after the Fukushima Daiichi nuclear disaster in March 2011. As a consequence, aerosol emissions from energy use in some regions may increase and therefore offset aerosol reductions elsewhere.".

**References**

Crippa, M., G. Janssens-Maenhout, F. Dentener, D. Guizzardi, K. Sindelarova, M. Muntean, R. Van Dingenen, and C. Granier (2016), Forty years of improvements in European air quality:

regional policy-industry interactions with global impacts, Atmospheric Chemistry and Physics, 16(6), 3825-3841.

Samset, B., M. Sand, C. Smith, S. Bauer, P. Forster, J. Fuglestvedt, S. Osprey, and C. F. Schleussner (2018), Climate impacts from a removal of anthropogenic aerosol emissions, Geophysical Research Letters, 45(2), 1020-1029.

Samset, B., G. Myhre, P. Forster, Ø. Hodnebrog, T. Andrews, G. Faluvegi, D. Flaeschner, M. Kasoar, V. Kharin, and A. Kirkevåg (2016), Fast and slow precipitation responses to individual climate forcers: A PDRMIP multimodel study, Geophysical Research Letters, 43(6), 2782-2791.

Shindell, D., and G. Faluvegi (2009), Climate response to regional radiative forcing during the twentieth century, Nature Geoscience, 2(4), 294.

Shindell, D., M. Schulz, Y. Ming, T. Takemura, G. Faluvegi, and V. Ramaswamy (2010), Spatial scales of climate response to inhomogeneous radiative forcing, Journal of Geophysical Research: Atmospheres, 115(D19).

---

## Author Comment (AC3) · 25 Oct 2019

We are grateful to the reviewer for their interest and comments on the paper. These comments are very valuable and have helped improve the manuscript. Here we outline how we have addressed these comments in the revised manuscript. The newly added discussions and rephrased sentences have been highlighted in green in our replies below.

The study uses the CESM global climate model to quantify the impact on effective radiative forcing, surface temperature, and precipitation rates of increases in aerosol emissions due to energy generation and decreases in emissions due to technological advances over the period 1970-2010.

The authors find that technological advances only partly offset impacts of energy generation. They also highlight the non-linearity of effective radiative forcing and diversity in temperature sensitivities. The paper is well written, and the figures support the discussion well. Figure 5 is perhaps the most novel aspect of the study and acts as a very efficient summary of the main findings. There are however results I do not understand, and the discussion could be improved in places. For these reasons, I recommend minor revisions.

While revising the manuscript, we realised that the isolation of the effects of aerosol changes in the "best estimate" experiment was probably biased due somehow to the experimental set-up. To address this issue, we have carried out a new experiment, which is now used throughout the revised manuscript (including Figures and texts). Note that, while this has resulted in different estimates of the impacts of aerosol changes in the best estimate case, it does not have any bearing on the major findings of this study on the comparison of the two different retrospective emission scenarios.

Main comment

1.       I understand why temperature sensitivities, as defined in the study, could be negative (i.e. located in the top left corner of Figure 5) locally because (i) albedo and circulation feedbacks mean that temperature responses are not collocated with ERF and (ii) both ERF and surface temperature distributions are inherently noisy. But I cannot understand how temperature sensitivities can be negative globally, as stated for the best estimate (Page 7 line 16). Is that really the case? According to Fig 3 and 4, ERF is $+0.1$ W m$^{-2}$ and DeltaT is $+0.09$, so the sensitivity should be positive. If there are negative global sensitivities, I would be concerned that the model does not conserve energy.

We are grateful to the reviewer for pointing this issue out. As mentioned above, we have accrued out a new model simulation to isolate the effects of aerosol changes in the "best estimates" experiment. As a result, we have repeated the related analysis and edited the manuscript accordingly (including Figures and texts).

The sign of global mean temperature response to aerosol changes isolated using the new experiment is now consistent with that of the ERF. There are still pronounced inconsistencies at regional scales, and a discussion of the underlying mechanisms has now been included in the revised manuscript.

Other comments

2.	Abstract, page 1, line 19: "must be interpreted in the context of experiment designs" is a cryptic statement that is in some way true of all experiments. I would suggest rewriting, perhaps to something like "must be interpreted in the context of the reference baseline".

By "the context of experiment designs" we refer not only to the reference baseline (i.e., background warming, concentrations of GHG and aerosol levels, and etc.), but also to those experiments where emissions/concentrations are scaled rather arbitrarily to maximise aerosol signals (e.g., the PDRMIP experiment), given the existence of large nonlinearities in the aerosol impact on climate (Feichter et al., 2004; Ming and Ramaswamy, 2009; Dobricic et al., 2019). As such, we decide to keep this statement as it is.

3.	Page 3, lines 3–4: That statement could be toned down – improving air quality is getting harder, and climate feedbacks (e.g. this year's record Arctic wildfires) may negate parts of the decreases.

We thank the reviewer for the suggestion, and have toned down the statement into "Anthropogenic aerosol emissions are expected to be reduced worldwide during the 21$^{st}$ century (Markandya et al., 2018)."

4.	Page 3, line 10: The reason why the impact of future changes in aerosols have not been looked in isolation before is that (i) uncertainties in emission pathways also affect other forcings, so aerosols are not alone in that respect, and (ii) considering all forcings together makes sense because sectors emit a cocktail of gaseous and particulate species. So the authors should make a better case for what we really gain at considering aerosols in isolation.

We agree with the reviewer regarding the links between aerosol emission pathways and those of other forcing agents, as well as the existence of large uncertainties in their future evolution. We also acknowledge that the nonlinear interactions between aerosols and other forcings are difficult to quantify. Nevertheless, we argue that it is still very useful to quantify the uncertainty range of future climate projections related to uncertainties in aerosol emission pathways, while all the other things being the same.

In response to the comment, we expand P3 L10 into "Yet, possible differences in the climate response to varying aerosol emissions trajectories, all the other forcings being the same, have been mostly overlooked so far (e.g., Sillmann et al. (2013); Pendergrass et al. (2015); Bartlett et al. (2016)). This, nevertheless, is still useful for partially assessing the uncertainty range of future climate projections related to uncertainties in aerosol-related emission pathways alone, despite the fact that emissions of GHGs also differ between those emission pathways."

5.  Page 3, lines 19–21: What did the Turnock study find?

We thank the reviewer for the suggestion. We have rephrased the statement into "Using a chemistry-climate model, Turnock et al. (2016) reported that the avoided aerosol/precursor emissions due to legislation and technology measures have improved air quality and human health over Europe, but have also led to a regional warming of up to 0.45 ±0.11 °C"

6.  Section 2.2: Which aerosol radiative forcing mechanisms does the model represent?

The CESM (CAM5) model we employed in this study uses the Modal Aerosol Module 3 (MAM3) which includes both aerosol direct (aerosol-radiation interactions) and indirect (aerosol-cloud interactions) effects.

7.  Page 5, line 5: permutations -> perturbations

Corrected

8.  Page 5, line 29: Technically speaking, a decrease of−4 is an increase!

We thank the reviewer for pointing this out; we have corrected the text accordingly.

Captions of Fig 2 and 3: "raised up" is unnecessarily ambiguous. Simply say "multiplied"

Done

9.  Page 6, line 15 Why is there an increase over Southern Africa? It looks statistically significant. Are there technological advances that increase emissions?

The reviewer is right in pointing out the statistically significant increases in AOD over Southern Africa in the technology advances experiment. We examined changes in each individual aerosol species, and found this is predominantly associated with sea-salt and organic aerosol emission changes in the technology advances experiment.

We added comments on this at the end of Section 3.1 as "It worth noting the AOD increases over the Southern Africa due to increases in sea-salt and OC which may be related to the additional warming induced changes in meteorology in the technology advances experiment."

10.  Page 6, lines 22–24: That statement is not obvious to me. The area of strongest cooling, in Siberia, looks to be to the north of where emissions changes are probably located, in China. (It would really help to see maps of emission changes.)

We thank the reviewer for the comment, and note again we used a new experiment to isolate the impacts of aerosol changes in the "best estimates" experiment. All related analysis has been updated, with the underlying mechanisms now included.

11.     Page 6, line 31: Is the enhancement again due to sea-ice albedo?

We tend to agree with the reviewer that the enhancement is mainly through the local sea-ice feedbacks. We have included analysis on the underlying mechanism in the revised Results section.

12.     Caption of Fig 5: "error bars represent model uncertainty" – how is that calculated?

The uncertainty is calculated as the $25^{th}$ -$75^{th}$ percentile spread of the differences between two sets of the 30 annual mean values. That is, the uncertainty is reported as the interquartile ($25^{th}$ -$75^{th}$ percentile) range of annual mean differences.

13.     Section 4.1: An alternative conclusion of that section is that trying to decompose ERFs and response into component-based estimates is misleading in an internal mixture context. Doing so is just not as informative as it first appears. Source-based decomposition would be more useful and, perhaps, less subject to nonlinearities.

We are grateful to the reviewer for providing a new angle to interpret our results. We assume that by "source-based" the reviewer means decomposing emissions into each individual emission sectors.  We agree that such decomposition may be less sensitive to nonlinearities, given that emission sectors emit a cocktail of gaseous and particulate species. However, note this is a different research problem from the one we are trying to address here. We designed the sensitivity experiments to investigate the differences between absorbing (mainly BC) and scattering (SO4) aerosols, and to assess the relative contribution of changes in each species to the total. We feel that the nonlinearities found here are very worth stressing given that massive efforts have been made to examine the climate impacts of different aerosol species separately (e.g., in the PDRMIP project (Myhre et al., 2017)).

14.     Page 11, line 11: "Despite the global mean negative forcing from 1970-2010 aerosol changes" – but according to Fig 3, your best estimate ERF is positive.

We apologize, there was an error in the figure. This has been corrected in the revised version.

15.     Page 11, line 21–24: Beyond model dependence, being critical of one's model is also useful. CAM is not the best-behaved model in Malavelle et al. 2017 doi:10.1038/nature22974, strongly overestimating rapid adjustments to aerosol-cloud interactions. But then even the best-behaved model gets the wrong rapid adjustments compared to observations Toll et al. 2017

doi:10.1002/2017GL075280. That would imply that ERFs, and the subsequent temperature and precipitation responses, are too strong in that model.

We thank the reviewer for the comment. We have expanded P11Ls21-24 into "We acknowledge the caveat/limitation of this study that all our findings may be model dependent, which is particularly the case for aerosols, given the high degree of parameterisation and divergence in aerosol schemes across present generation climate models. Also note carefully again that CAM5 has relatively larger aerosol forcing compared to other Coupled Model Intercomparison Project Phase 5 (CMIP5) models (Allen and Ajoku, 2016; Malavelle et al., 2017; Toll et al., 2017; Zhou and Penner, 2017). "

**References**

Allen, R. J., and Ajoku, O.: Future aerosol reductions and widening of the northern tropical belt, Journal of Geophysical Research: Atmospheres, 121, 6765-6786, 2016.

Bartlett, Bollasina, M. A., Booth, B. B., Dunstone, N. J., Marenco, F., Messori, G., and Bernie, D. J.: Do differences in future sulfate emission pathways matter for near-term climate? A case study for the Asian monsoon, AGU Fall Meeting Abstracts, 2016,

Dobricic, S., Pozzoli, L., Vignati, E., Van Dingenen, R., Wilson, J., Russo, S., and Klimont, Z.: Nonlinear impacts of future anthropogenic aerosol emissions on Arctic warming, Environmental Research Letters, 14, 034009, 2019.

Feichter, J., Roeckner, E., Lohmann, U., and Liepert, B.: Nonlinear aspects of the climate response to greenhouse gas and aerosol forcing, Journal of climate, 17, 2384-2398, 2004.

Malavelle, F. F., Haywood, J. M., Jones, A., Gettelman, A., Clarisse, L., Bauduin, S., Allan, R. P., Karset, I. H. H., Kristjánsson, J. E., and Oreopoulos, L.: Strong constraints on aerosol–cloud interactions from volcanic eruptions, Nature, 546, 485, 2017.

Ming, Y., and Ramaswamy, V.: Nonlinear climate and hydrological responses to aerosol effects, Journal of Climate, 22, 1329-1339, 2009.

Myhre, Forster, P., Samset, B., Hodnebrog, Ø., Sillmann, J., Aalbergsjø, S., Andrews, T., Boucher, O., Faluvegi, G., and Fläschner, D.: PDRMIP: A precipitation driver and response model intercomparison project—Protocol and preliminary results, Bulletin of the American Meteorological Society, 98, 1185-1198, 2017.

Pendergrass, A. G., Lehner, F., Sanderson, B. M., and Xu, Y.: Does extreme precipitation intensity depend on the emissions scenario?, Geophysical Research Letters, 42, 8767-8774, 2015.

Sillmann, J., Pozzoli, L., Vignati, E., Kloster, S., and Feichter, J.: Aerosol effect on climate extremes in Europe under different future scenarios, Geophysical Research Letters, 40, 2290-2295, 2013.

Toll, V., Christensen, M., Gassó, S., and Bellouin, N.: Volcano and Ship Tracks Indicate Excessive Aerosol‐Induced Cloud Water Increases in a Climate Model, Geophysical research letters, 44, 12,492-412,500, 2017.

Zhou, C., and Penner, J. E.: Why do general circulation models overestimate the aerosol cloud lifetime effect? A case study comparing CAM5 and a CRM, Atmospheric Chemistry and Physics, 17, 21-29, 2017.

---

## Author Comment (AC4) · 25 Oct 2019

We are grateful to the reviewer for their interest and comments on the paper. These comments are very valuable and have helped improve the manuscript. Here we outline how we have addressed these comments in the revised manuscript. The newly added discussions and rephrased sentences have been highlighted in green in our replies below.

This paper assesses the influence on radiative forcing, temperature and precipitation of changes in anthropogenic aerosol emissions between 1970 and 2010 in the CESM1-CAM5 model. The paper decomposes the effects into the contribution from aerosol emissions increases due to increased energy use and the contribution form aerosol emissions decreases due to technological advances and air quality abatement initiatives over the same time period. The authors find that the two effects have partially cancelled each other over the analyzed period and do so nonlinearly and non-uniformly. The paper is well written overall. There are several useful and interesting points made. The experiment design is clean and well thought out. However, I find that the motivation and certain aspects of the analysis require additional depth, in order to make the novelty of the paper clearer. I am also concerned that certain aspects of the results seem counterintuitive, suggesting problems with the model set-up (see Other Major Comments 2 and 3 below). At present it is not entirely clear what new the authors are adding to the conversation, though I do believe that this analysis has the potential to be a useful contribution with some additional depth. I believe that major revisions to provide further depth would be needed for the paper to merit publication in ACP. These are discussed further below:

While revising the manuscript, we realised that the isolation of the effects of aerosol changes in the "best estimate" experiment was probably biased due somehow to the experimental set-up. To address this issue, we have carried out a new experiment, which is now used throughout the revised manuscript (including Figures and texts). Note that, while this has resulted in different estimates of the impacts of aerosol changes in the best estimate case, it does not have any bearing on the major findings of this study on the comparison of the two different retrospective emission scenarios.

1.      The best estimate results shown in this manuscript are essentially an assessment of impact of aerosols on climate over the last 40 years, which has been conducted extensively in the existing literature. The main novelty in the paper, therefore, lies in the decomposition into the two factors described above. However, I believe the authors must do more to center and frame this decomposition.

At present, the authors do not make a strong case for why decomposing the total effect into these two components is a valuable exercise. This should be explicitly laid out in the introduction.

The authors need to be clearer that the assessment of last 20th century effects of aerosols has been widely done elsewhere and provide this more as background, acknowledging where their work is replicative of and consistent with that done elsewhere, and focus more thoroughly on unique insight gained through the de-composition.

As the reviewer points out, the main contribution of this work is the analysis of the climate changes associated with the two aerosol emission drivers, but not the historical aerosol changes

(referred to as "best estimate" in this work). Note however that one significant difference with most of the previous studies is that, even in the "best estimate" scenario, we analyse equilibrium rather than transient responses. Also, the "best estimate" experiment is thoroughly analysed in a manuscript that is under review for the *Journal of Geophysical Research: Atmosphere* (Zhao et al. Climate forcing and response to greenhouse gases, aerosols and ozone in CESM1), in which the impacts of historical aerosol changes have been explicitly discussed in the context of existing works. The ERF, as well as temperature and precipitation responses to the historical aerosol emission changes are presented here primarily to provide a reference for the two aerosol emission scenarios.

To clarify the motivation of this work, we expand P3 L22-24 into "As discussed above, energy use growth and technology advances are two of the major policy-relevant drivers of past aerosol/precursor emission changes via, for example, changes in power generation, industry and transportation. These drivers are very likely to continue to play important but competing roles in modulating future aerosol emission, as we gradually transit to a new energy structure. An analysis of the climate impact to recent changes in the two above emissions drivers is therefore critically important for future aerosol related climate projections and climate change impact reduction strategies. Here we perform time-slice model simulations using the fully-coupled Community Earth System Model (CESM1), seeking to quantify the climate forcing and impacts of aerosol changes related two the above policy-relevant emission drivers (energy use growth and technology advances) at both global and regional scales."

At the same time, the analysis of the decomposition should be further deepened. Although the authors report many signals, they do not provide any depth of analysis on why these signals emerge. Given how many other studies have analysed the role of aerosol increases and decreases in the second half of the 20th century and made many of the same points the authors make here, simply reporting these signals is not sufficient. Where do the two effects counteract or reinforce each other in ERF, temperature, or precipitation? Why does this occur? What are the implications for future anticipated shifts in aerosol emissions due to the two effects?

Regarding the implications for future anticipated shifts in aerosol emissions, please refer to our replies to comment #6.

Also, we have added comments on changes in low-level circulation and sea level pressure (Figure S2 in the supplement), trying to provide some speculations around the underpinning mechanisms of temperature/precipitation responses. Please refer to the new result section n the revised manuscript. However, although it is certainly interesting and relevant to shed light on

the underpinning mechanisms, we feel strongly that such analyses are speculative and with large uncertainties. Therefore, in the revised manuscript, we tried not to inflate the revised manuscript with mechanism analysis, in order also not to overwhelm the main storyline of this study which is around the climate responses to the two policy-relevant drives.

2.      A main result that the authors emphasize is the result that the individual ERFs of the aerosol species do not add up linearly, giving rise to the authors' repeated statement that ERF calculations should be considered to be dependent on experiment set-up. While this is in some ways a fairly obvious statement, it would be useful to have this explored in further detail. However, the authors do not provide any explanation for this result. I believe the authors must assess the mechanism behind this more thoroughly for it to be a valuable contribution. Is the nonlinearity of the ERF when the aerosols are emitted separately versus together a result of changes in the total atmospheric burden or of changes in the spatial distribution and resulting radiative interactions? The authors allude briefly to the total burdens of each species being "identical" in the separate and combined simulations (P8, L20), but do not show it. This should be explicitly shown. The remaining explanations that are currently only suggested in Section 4.1 should be explicitly assessed.

We thank the reviewer for the valuable comments and suggestions.

Both spatial patterns and the global mean values of aerosol burden changes do not show appreciable differences between the two different experiments. This is consistent for all the three species (Figure R4.1). However, despite identical changes in aerosol burdens between the two different experiments, the diagnosed ERF do not add up, leading us to stress the nonlinearity in estimating the ERF. To make the statement clearer, we rephrase P8 L30-31 as "We note that changes in both the spatial pattern and the global mean amount (Figure S3) of the burden and AOD of the three aerosol species do not show appreciable differences to those in the experiment where all the three species change simultaneously (B10-B70)."

We also thank the reviewer for suggesting us to assess and expand the remaining discussion in Section 4.1. While this is certainly relevant, we believe such additional experiments (e.g., having sensitivity experiments where the amounts of background soluble aerosol species vary while keeping BC constant) will make the scope of this work too wide and the manuscript too heavy. Nevertheless, in combination with comment #2 from reviewer #2, we expanded the discussions into "This reflects partly the nonlinear effect associated with the mixing state of different aerosol species as well as the importance of background aerosol loadings. This is particularly important for BC whose effects depend also on the presence of sulphate and organic aerosols (Ramana et al., 2010). That is, given that aerosol species are internally-mixed in MAM3 (i.e. different chemical species are mixed within an aerosol particle), the

hygroscopicity of aerosol particles is dominated by the volume of soluble species (organic compounds and sulphate). This means that the nonlinearity in the isolated aerosol ERF may be a reflection of the aerosol scheme in CESM1. More specifically, BC particles tend to be coated with other species during ageing, thereby enhancing the absorption effects and the subsequent impacts on cloud microphysics, as well as amplifying their radiative forcing (Haywood and Ramaswamy, 1998; Kim et al., 2008; Chung et al., 2012; Wu et al., 2016). …"

Also, we further tone down the statement in P9 L14-16 into "Overall, the above discussion illustrates the importance of background aerosol concentrations in estimating the radiative forcing of aerosols. For example, we speculate that diagnosing the ERF of BC the other way round, namely, keeping all other aerosol species at 1970 levels while changing BC to 2010 levels would likely result in a different ERF estimate."

[Figure]

Figure R4.1 The 1970-2010 changes in the burden (mg m$^{-2}$) of black carbon (BC, top row), organic carbon (OC, second row) and sulphate (SO$_4$, third row) in the best estimate experiment.

The left column is from the experiment where all aerosol species change simultaneously, while the right column is for those isolated between the reference 2010 run and the one where the emissions of a targeted species (e.g. BC in the top right) are fixed to its 1970 levels.

Other Major Comments:

3.      Confidence intervals should be provided for all global-mean values.

Done

4.      I agree with Anonymous Referee 3 that the negative global mean temperature sensitivity value in the best estimate case seems to violate basic energy conservation. How can a positive global-mean ERF result in a negative global-mean temperature change or vice versa? This suggest a major issue with the model formulation. If it is somehow robust, it needs to be explained.

As mentioned above (also see our reply to reviewer #3), we added a new model simulation to isolate the effects of aerosol changes in the "best estimates" experiment. As such, we repeated all related analysis and edited the manuscript accordingly (including Figures and texts).

The sign of global mean temperature response to aerosol changes isolated using the new experiment now consistent with that of ERF. However, there are still pronounced inconsistencies at regional scales, and the underlying mechanism has been thoroughly discussed in the revised manuscript.

5.      Figure 1: Why does the large increase in SO$_4$ burden result in a positive ERF? This needs to be explained, particularly when compared to the larger negative ERF of the smaller (and less reflective) OC burden.

The positive global mean SO$_4$ ERF, as we stated in P6 Ls1-3, is due to the partial cancellation between the pronounced positive forcing from sulphate aerosol reductions over Europe and North America and the relatively confined negative forcing from sulphate aerosol increases over Asia (Figure 1f). The former is amplified over the Arctic, and results into a net global mean positive forcing.

The larger negative OC ERF explainable, since the small OC reductions over Europe and North America leads to only slight positive OC ERF (Figure 1e).

In response to this comment, we expand P6 Ls1-3 into "The global mean ERF of sulphate aerosols is small and positive, because of the partial cancellation between the negative forcing from sulphate aerosol increases over Asia and the pronounced positive forcing from sulphate aerosol reductions over Europe and North America which is amplified over the Arctic (Fig. 1f)."

6.      Section 4.3 is largely a literature review on the formulation of aerosol emissions scenarios and does not explore what implications the results of this analysis have for future projections. How may their findings "help better assess and interpret such uncertainties in future climate projections"?

We thank the reviewer for these comments.

In combination with comment#1 and comments from the anonymous reviewers #2 and #3, we have expanded Section 4.3 into "Reliable projections of future climate under different but equally plausible emission pathways are of utmost importance to better constrain the range of possible societal risks and response options. Unfortunately, there are still considerable challenges due to limitations and uncertainties in our understanding of many aspects of the climate system (Knutti and Sedláček, 2013; Northrop and Chandler, 2014; Marotzke, 2019). Aerosols represent one of the largest sources of uncertainty (Boucher et al., 2013; Lee et al., 2016; Fletcher et al., 2018). Present-day anthropogenic aerosol emissions are largely influenced by sectors including power generation, industry and transport. However, in some of the future emission pathways, for example, the Tier-1 Shared Socioeconomic Pathways scenarios (SSP1; Gidden et al. (2018)), aerosol emissions are expected to decline drastically worldwide as we transit to non-fossil-fuel-based fields together with rapid implementation of air pollution control measures and new technologies. For example, mainly as a result of China's transition to a less energy-demanding society, for the first time the global coal consumption decreased in 2015 since the 1970s (Worl Energy Council, 2016). However, the timing and rate of such transitions are largely uncertain. On the other hand, it is also likely that aerosol-related emissions will increase, especially over some developing regions, under scenarios where high inequality exists between and within countries. For example, in SSP3, expanding industrial sectors over Southeast Asia may continue to rely on traditional energy sources such as coal for much of the 21$^{st}$ century. Also, it is possible that the world may continue to rely on fossil energy sources more strongly than expected over the coming years, given the concerns about nuclear energy after the Fukushima Daiichi nuclear disaster in March 2011. As a consequence, aerosol emissions from energy use in some regions may increase and therefore offset aerosol reductions elsewhere.

The above discussion reflects the large uncertainties (both spatially and temporally) in our understanding and estimates of future aerosol-related emission trajectories, given the possibility that very different future emission pathways may be adopted by different countries to compromise between climate/air pollution impacts and economic growth. Our findings may

help better assess and interpret such uncertainties in future climate projections associated with changes in aerosols. First of all, the large impacts of present-day aerosol emissions from the two competing drivers, as reported in this work, suggest that the major drivers (e.g., future energy structure and efficiency, air pollution control measurements, as well as technology progresses) of aerosol emission changes are likely to continue to play important roles in future climate projections. Secondly, uncertainties in future aerosol emissions pathways combine with those of other climate forcing agents (e.g., greenhouse gas emissions and land-use changes). Such uncertainties influence the impacts of aerosol forcing through changing the background climate state (see Section 4.2; e.g.,(Frey et al., 2017; Nordling et al., 2019; Stolpe et al., 2019). More importantly, our results stress the importance of nonlinearities when comparing and assessing the impacts of different future aerosol emission trajectories. This adds further caveats in interpreting future climate projections related to aerosol changes in addition to uncertainties in emission pathways of both aerosols and their precursors and GHGs."

Minor Comments:

7.      The authors use a somewhat confusing formulation throughout the paper in which they refer to both the aerosol increases from increased energy use and the aerosol decreases from technological advances as "aerosol emissions" (e.g. P8, L23; P10, L26). I recommend rewording these to something like "aerosol emissions changes" throughout, to make clearer that it is in some case the absence rather than the presence of aerosol emissions that is being analysed.

We thank the reviewer for the suggestion, and have thoroughly edited the manuscript.

8.      The model and experiment design section would benefit from a clearer description in the text of what the different model simulations are and what scenarios they are intended to test. This is provided at the moment entirely in the caption to Table 1, but should be at least summarized in Section 2.2.

We thank the reviewer for this suggestion. We have rephrased P5 Ls1-4 into "The baseline 2010 experiment (B10) was initialised using the year 2010 model dump from one ensemble member (No 34) of the CESM1 large ensemble (Kay et al. (2015) transient historical experiment, and was driven by the 2010 all forcing factors (Table 1). Also, we have three perturbation experiments where anthropogenic aerosols are perturbed using different emission scenarios (i.e., the 1970 best estimate, STAG_ENE and STA_TECH as described in Sect. 2.1) while all others forcing agents (e.g., GHGs, natural aerosols, land use, solar forcing) are the same as in the B10 run, in order to differentiate the impacts of the two aerosol emission drivers (refer Table 1 for more details)."

9.      It is not entirely clear what the residual emissions are that are occurring in the B10 simulation but that are not captured in the SEN and STC simulations. It would help for this to be more thoroughly described and for its magnitude relative to the emissions changes to be provided, especially given the authors repeated references to it.

We agree with the reviewer that such information would be beneficial. Note, however, that the two retrospective emission scenarios were designed in a way that does not provide a linear decomposition of the total 1970-2010 emission changes. As such, it is impossible for us to quantify the residuals. To make the point clear, we added a sentence at the end of Section 2.1 to direct readers to Crippa et al. (2016) "For more details regarding the nonlinearity associated with the retrospective emission scenarios, please refer to Crippa et al. (2016)."

10.     P3, L3-4: recommend replacing "will" with "may", as certain of the Shared Socioeconomic Pathways do simulate regional increases in anthropogenic aerosol emissions.

In combination with comment #3 from reviewer #3, we tone down the statement into "Anthropogenic aerosol emissions are expected to be reduced worldwide during the 21$^{st}$ century (Markandya et al., 2018)."

11.     P5, L5: "where the climate system equilibrates to imposed permutations but the deep ocean" ←this phrase is unclear

The sentence has been rewritten into "where the surface climate system equilibrates to imposed perturbations; NB the deep ocean may take longer to equilibrate"

12.     P7, L30: The drying over Europe as well as Asia seems counterintuitive, since all the other analysis suggested opposite trends (in burden, ERF, and temperature) over Europe and Asia. This bears further explanation.

We apologise for misleading the reviewer to a wrong interpretation. Note our results agree with previous studies in producing a wettening over Europe and a drying over Asia in the best-estimate emission scenario (Figure 6a).

L30 here is for the "energy use" experiment where aerosols increase globally. To make the statements even clearer, we rephrase P7 Ls28-30 into "The globe, especially land areas, gets drier in response to aerosol emission changes from energy use growth (Fig. 6b). The precipitation change in Asia (-0.11±0.30 mm day$^{-1}$) is close to that associated with the best estimate of 1970-2010 aerosol emission changes (-0.13±0.28 mm day$^{-1}$)."

13.     P8, L7-9: I'm fairly certain the estimate from this paper and from PDRMIP have almost entirely overlapping error bounds, making this difference hardly worth noting. I would rather

read this as suggesting that the precipitation sensitivity is very well aligned with that found by PDRMIP. This is one reason why the authors need to provide confidence intervals on their global-mean values.

First, note our new model experiment to isolate the effects of aerosol changes in the "best estimates" scenario.

P8 Ls7-8 has been rewritten into "Generally, precipitation changes with temperature at a rate of 0.09-0.15 mm day$^{-1}$ K$^{-1}$. This is slightly larger than the estimate (~28.6 mm yr$^{-1}$ K-1, i.e., ~0.08 mm day-1 K-1) for the slow climate response component derived from the Precipitation Driver Response Model Intercomparison Project (PDRMIP; Samset et al. (2016))."

14.     P8, L10-12: "This may suggest that regional.." This statement seems deeply obvious and well-established in the community. Recommend cutting.

We tend to agree with the reviewer that this is "well-established", but we still feel necessary to stress the point here with our results.

We tone down P8Ls10-12 into "This supports previous studies demonstrating that regional precipitation responses are not simply linked to temperature through energy budget constraints, but also depend on other factors such as adjustments in the atmospheric circulation and remote teleconnections (Bollasina et al., 2014; Wilcox et al., 2018; Lewinschal et al., 2019)"

15.     Figure 1: It is not clear that the presence of hatching means that it is statistically significant rather than no statistically significant until one reads the text. Recommend rewording L5-6 to be clearer. It is also quick difficult to see the statistical significance hatching over the ERF values – recommend a different colour (e.g. light grey) for the hatch lines.

To make the hatches more readable, we change the black "//" hatches into grey "//" in Figure 1,2,4 and 6.

Ls5-6 are rewritten into "The statistical significance at 5% level is calculated using the two-tailed student t-test and is denoted by the grey hatches."

16.     Figure 5: It is incredibly difficult to visually parse the symbols. The Global and Global Ocean symbols are essentially indistinguishable in the crowded areas of the graph. Very few of these are actually discussed in the manuscript, and I recommend removing some of the symbols (perhaps the latitude band symbols) to make this figure legible.

We appreciate that Figure 5 is not easy to read, but feel necessary to keep the latitude band values to support our statement in P7 L14.

We have updated Figure 5 with the new experiment included, making it much more readiable thatn before.

Typographic comments:

17.     There are several instances of in-line citations still being enclosed by parentheses that should be corrected (P4: L27, L28, L31)

Corrected.

18.     P7, L13: "domain "→"domains"

Corrected.

19.     P9, L12-13: "is difficult to be quantified "→"are difficult to quantify"

Corrected.

20.     The paper contains a couple stub sentences without verbs: p4, L14-15 and p8, L27-28

Corrected.

References

Boucher, O., Randall, D., Artaxo, P., Bretherton, C., Feingold, G., Forster, P., Kerminen, V.-M., Kondo, Y., Liao, H., and Lohmann, U.: Clouds and aerosols, in: Climate change 2013: the physical science basis. Contribution of Working Group I to the Fifth Assessment Report of the Intergovernmental Panel on Climate Change, Cambridge University Press, 571-657, 2013.

Crippa, M., Janssens-Maenhout, G., Dentener, F., Guizzardi, D., Sindelarova, K., Muntean, M., Van Dingenen, R., and Granier, C.: Forty years of improvements in European air quality: regional policy-industry interactions with global impacts, Atmospheric Chemistry and Physics, 16, 3825-3841, 2016.

Fletcher, C. G., Kravitz, B., and Badawy, B.: Quantifying uncertainty from aerosol and atmospheric parameters and their impact on climate sensitivity, Atmospheric Chemistry and Physics, 18, 17529-17543, 2018.

Frey, L., Bender, F. A.-M., and Svensson, G.: Cloud albedo changes in response to anthropogenic sulfate and non-sulfate aerosol forcings in CMIP5 models, Atmospheric Chemistry and Physics, 17, 9145-9162, 2017.

Gidden, M., Riahi, K., Smith, S., Fujimori, S., Luderer, G., Kriegler, E., van Vuuren, D. P., van den Berg, M., Feng, L., and Klein, D.: Global emissions pathways under different socioeconomic scenarios for use in CMIP6: a dataset of harmonized emissions trajectories through the end of the century, Geoscientific Model Development Discussions, 1-42, 2018.

Kay, J., Deser, C., Phillips, A., Mai, A., Hannay, C., Strand, G., Arblaster, J., Bates, S., Danabasoglu, G., and Edwards, J.: The Community Earth System Model (CESM) large ensemble project: A community resource for studying climate change in the presence of internal climate variability, Bulletin of the American Meteorological Society, 96, 1333-1349, 2015.

Knutti, R., and Sedláček, J.: Robustness and uncertainties in the new CMIP5 climate model projections, Nature Climate Change, 3, 369, 2013.

Lee, L. A., Reddington, C. L., and Carslaw, K. S.: On the relationship between aerosol model uncertainty and radiative forcing uncertainty, Proceedings of the National Academy of Sciences, 113, 5820-5827, 2016.

Marotzke, J.: Quantifying the irreducible uncertainty in near‐term climate projections, Wiley Interdisciplinary Reviews: Climate Change, 10, e563, 2019.

Nordling, K., Korhonen, H., Räisänen, P., Alper, M. E., Uotila, P., O'Donnell, D., and Merikanto, J.: Role of climate model dynamics in estimated climate responses to anthropogenic aerosols, Atmospheric Chemistry and Physics, 19, 9969-9987, 2019.

Northrop, P. J., and Chandler, R. E.: Quantifying sources of uncertainty in projections of future climate, Journal of Climate, 27, 8793-8808, 2014.

Samset, B., Myhre, G., Forster, P., Hodnebrog, Ø., Andrews, T., Faluvegi, G., Flaeschner, D., Kasoar, M., Kharin, V., and Kirkevåg, A.: Fast and slow precipitation responses to individual climate forcers: A PDRMIP multimodel study, Geophysical Research Letters, 43, 2782-2791, 2016.

Stolpe, M. B., Medhaug, I., Beyerle, U., and Knutti, R.: Weak dependence of future global mean warming on the background climate state, Climate Dynamics, 1-21, 2019.

Worl Energy Council: World energy resources 2016, World Energy Council, London, UK, 2016.

---

## Author Comment (AC5) · 25 Oct 2019

Dear Editor and referees,

Many thanks for your comments that have helped improve this manuscript substanially.

While revising the manuscript, we realised that the isolation of the effects of aerosol changes in the "Best Estimate" experiment was probably biased due somehow to the experimental set-up (related to the initial condition issue). To address this issue, we have carried out a new experiment, which is now used throughout the revised manuscript (including Figures and texts). Note that, while this has resulted in differ-

ent estimates of the impacts of aerosol changes in the best estimate case, it does not have any bearing on the major findings of this study on the comparison of the two different retrospective emission scenarios.

On behalf of all co-authors and sincerely,

Alcide Zhao

alcide.zhao@ed.ac.uk